# Diagnosing and Mitigating Modality Interference in Multimodal Large Language Models

## Abstract

Multimodal Large Language Models (MLLMs) have demonstrated impressive capabilities across tasks, yet they often exhibit difficulty in distinguishing task-relevant from irrelevant signals—particularly in tasks like Visual Question Answering (VQA)—which can lead to susceptibility to misleading or spurious inputs. We refer to this broader limitation as the Cross-Modality Competency Problem—the model's inability to fairly evaluate all modalities. This vulnerability becomes more evident in modality-specific tasks—such as image classification or pure text question answering—where models are expected to rely solely on one modality. In such tasks, spurious information from irrelevant modalities often lead to significant performance degradation. We refer to this failure as Modality Interference, which serves as a concrete and measurable instance of the cross-modality competency problem, and we further design a perturbation-based causal diagnostic experiment to verify and quantify this problem. To mitigate modality interference, we propose a novel framework to finetune MLLMs, including perturbation-based data augmentations with both heuristic perturbations and adversarial perturbations, and a consistency regularization strategy applying on model outputs with original and perturbed inputs. Experiments on multiple benchmark datasets (image-heavy, text-heavy and multimodal tasks) and multiple model families with different scales demonstrate significant improvements in robustness and cross-modality competency, indicating our method's effectiveness in boosting unimodal reasoning ability while enhancing performance on multimodal tasks.

## 1 Introduction

Multimodal Large Language Models (MLLMs) have made significant strides in integrating vision and language understanding within a unified architecture (Liu et al., 2023b; Luo et al., 2023; Bai et al., 2025). By combining powerful visual encoders and large language models through alignment mechanisms, MLLMs such as LLaVA (Liu et al., 2023b) and Qwen-VL (Bai et al., 2025) demonstrate strong capabilities across a wide range of multimodal tasks. However, beneath their seemingly impressive performance lies a critical limitation: MLLMs often fail to distinguish between relevant and irrelevant signals across modalities, leading to unreliable predictions (Wang et al., 2024a; Zhu et al., 2024; Hosseini et al., 2025). Moreover, while MLLMs are designed for multimodal tasks, their failure on unimodal tasks—where only a single modality (e.g. text) should guide the prediction—raises concerns about whether the model can preserve modality-specific competencies. For instance, MLLMs frequently underperform on pure visual recognition (Zhang et al., 2024; Tong et al., 2024) and textual reasoning (Zhu et al., 2024; Wang et al., 2023; Lin et al., 2024), suggesting that cross-modal fusion may induce unintended interference and degrade unimodal performance.

Recent studies have attributed this phenomenon to a variety of symptoms arising during the vision-language alignment process, such as catastrophic forgetting (Zhang et al., 2024; Tong et al., 2024; Wang et al., 2023; Lin et al., 2024), knowledge conflict (Wang et al., 2024a; Zhu et al., 2024), and spurious correlations (Chen et al., 2024a; Hosseini et al., 2025). Catastrophic forgetting has been identified as a key factor in visual degradation (Zhang et al., 2024; Tong et al., 2024), where multimodal tuning of MLLM overrides its pretrained visual features. Cross-modal knowledge conflict (Wang et al., 2024a; Zhu et al., 2024) impairs pure-text reasoning, as models often produce

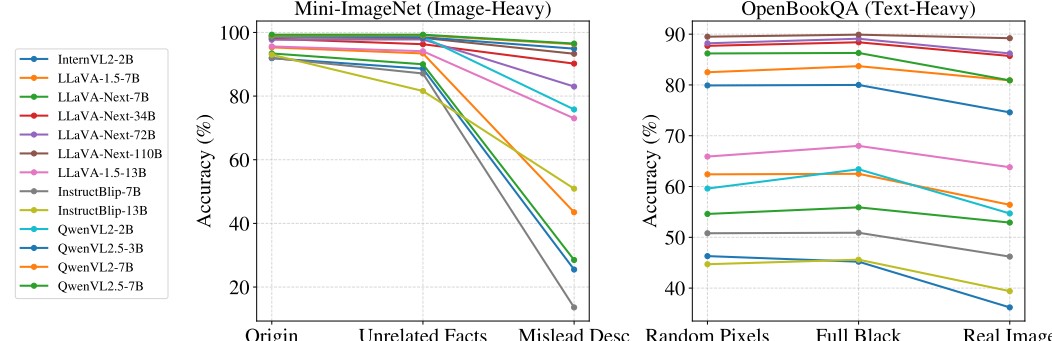

Figure 1: Performance degradation under irrelevant perturbations reveals modality interference in MLLMs. Left: Mini-ImageNet (image-heavy) with Original input, Unrelated Facts, and Misleading Descriptions. Right: OpenBookQA (text-heavy) with Random Pixels, Full Black Canvas, and Irrelevant Real Images. Misleading descriptions induces the most severe degradation in image-heavy tasks, while irrelevant real images cause the largest drop in text-heavy reasoning.

inconsistent outputs when visual inputs are introduced, reflecting misaligned visual and textual parametric memories. Additionally, studies on spurious correlations (Chen et al., 2024a; Hosseini et al., 2025; Zhou et al., 2025) show that MLLMs tend to rely on superficial cross-modal cues rather than task-relevant grounding. While these symptoms shed light on MLLMs' limitations, most works treat these issues in isolation. For instance, architectural issues such as shallow cross-modal fusion have been widely discussed: lightweight projectors in models like LLaVA (Liu et al., 2023b) fail to fully align vision and language representations, resulting in unstable modality reliance (Tong et al., 2024; Zhu et al., 2024; Zhao et al., 2025). Others attribute performance bottlenecks to data limitations—insufficient modality-specific supervision leads to impaired visual decoding (Zhang et al., 2024) and diminished language understanding (Lin et al., 2024). Inspired by these observations, our insight is to unify these challenges under a broader perspective: the model's inability to identify and rely on the modality that contributes most relevant information to the task. We argue that the fundamental limitation lies in MLLMs' lack of cross-modality competency (Gardner et al., 2021)—the ability to fairly evaluate and integrate information across modalities. Current MLLMs lack mechanisms to support this competency during inference, making them vulnerable to misleading cross-modal signals—a failure mode we refer to as **Modality Interference**.

To systematically diagnose and mitigate modality interference, we introduce a two-stage methodology grounded in causal analysis. First, we design a perturbation-based evaluation experiment inspired by causal intervention principles (Pearl, 1995; Chen et al., 2024a) to diagnose the extent of modality interference across tasks and model scales. Second, we propose a robust fine-tuning framework to mitigate modality interference. Specifically, in our evaluation analysis, we first focus on modality-heavy settings using a multiple-choice question answering format, where the model selects an answer from a fixed set of options based on both image and text input. We then include image-heavy tasks (e.g., image classification), text-heavy tasks (e.g., pure-text QA), and balanced multimodal tasks (e.g., VQA), allowing us to examine how models behave under different modality-reliance scenarios. To further induce modality interference, we introduce heuristic perturbations: In image-heavy tasks, we perturb the text input by prepending either (i) unrelated scientific facts or (ii) misleading descriptions that falsely associate an incorrect option with the image content. In text-heavy tasks, where the default visual input is random noise, we perturb the visual input with (i) semantically meaningful real images, (ii) full black canvas, or (iii) full white canvas. These perturbations are designed to either introduce spurious cues or reinforce irrelevant modality signals. We evaluate the resulting changes in model predictions to assess the robustness of modality selectivity. While the perturbation-based evaluations offer empirical insights, we further frame our analysis through a **causal intervention framework** and in which we model modality interference through a causal graph abstraction. Building on this framework, we evaluate a range of pretrained MLLMs across different architectures and scales with results shown in Fig. 1. In image-heavy tasks, unrelated textual facts moderately reduce performance, while misleading descriptions cause severe degradation—revealing the model's vulnerability to spurious textual cues. In text-heavy tasks, canvas inputs have little effect, but unrelated real images mostly hurt performance, indicating improper fusion of irrelevant visual signals into reasoning.

The empirical results from Fig. 1 confirm the presence of modality interference and reveal the limitations of current MLLMs in lack of cross-modality competency. To mitigate modality interference, we propose a perturbation-based fine-tuning framework for MLLMs. Specifically, to alleviate data insufficiency, we apply a perturbation-based data augmentation strategy, where we construct a diverse mixture of modality-specific, perturbation-augmented samples and original VQA samples. The perturbations include both heuristic variants (e.g., injecting unrelated facts into image-heavy prompts) and adversarial training-time perturbations, which expose the model to worst-case alignment disruptions and thus serve as a stronger form of regularization. To further improve robustness, we introduce a consistency regularization strategy(e.g., via Jensen–Shannon divergence), which enforces output stability between original and perturbed samples. In summary, the main contributions of this paper are threefold. First, we introduce the notion of the Cross-Modality Competency Problem to describe how multimodal models may struggle to balance different modalities, and analyze modality interference as one concrete instance of such challenges in MLLMs. Second, we design a perturbation-based causal evaluation experiment that systematically quantifies modality reliance and reveals models' susceptibility to modality interference. Third, we propose a fine-tuning strategy that combines supervised augmentation with both heuristic and adversarial perturbations and consistency regularization to mitigate modality interference. Extensive experiments across multiple MLLM families and diverse benchmarks demonstrate the superiority of our method.

## 2 RELATED WORKS

**Improving Modality Alignment in Multimodal Language Models** Recent studies have revealed that modality misalignment remains a key obstacle in MLLMs, leading to degraded performance on both image-heavy and text-heavy tasks. For visual understanding, catastrophic forgetting occurs when multimodal tuning overrides pretrained visual features (Zhang et al., 2024; Tong et al., 2024; Wang et al., 2023). In text-heavy scenarios, knowledge conflict (Zhu et al., 2024) arises when inconsistent parametric knowledge from different modalities confuse reasoning. Moreover, mDPO (Wang et al., 2024a) identifies language bias in training, where models fail to condition their responses on visual input. Some works attribute such issues to shallow fusion (Wang et al., 2023)—e.g., LLaVA uses lightweight projectors to bridge vision and language spaces, leaving a representational gap and resulting in loosely coupled features (Tong et al., 2024; Zhu et al., 2024; Zhao et al., 2025). Others highlight data limitations: even well-encoded visual features fail to support reasoning without adequate supervision to guide decoding (Zhang et al., 2024).

Building on these diagnoses, recent models have proposed multiple solutions. MoF (Tong et al., 2024) mitigates this by fusing features from multiple vision encoders, while VLMClassifier (Zhang et al., 2024) enhances recognition via vision-only finetuning, though it struggles with VQA due to lack of cross-modal alignment. CogVLM (Wang et al., 2023) introduces a visual expert module to improve vision-language integration. VILA (Lin et al., 2024), QwenVL (Bai et al., 2025), and InternVL (Zhu et al., 2025) incorporate text-only supervision in different ways to preserve or enhance language capabilities during multimodal training—through stage-wise separation, parallel preservation, and unified joint optimization, respectively. Similar patterns arise in multimodal structural reasoning, where models must rely on relevant modalities to generalize to unseen relations (Cai et al., 2024). These works motivate us to hypothesize on the root cause—the model's inability to assess modality relevance. We further propose a causal framing of *modality interference* and introduce a perturbation-based fine-tuning strategy to improve the inference-time robustness of MLLMs.

**Adversarial Robustness Across Modalities** Adversarial perturbations threaten the reliability of both vision and text tasks by exposing vulnerabilities through deliberate and imperceptible perturbations. Existing robustness methods can be broadly categorized by modality, targeting either continuous image embeddings or discrete token spaces. In vision tasks, attacks like FGSM (Goodfellow et al., 2015) and CW (Carlini & Wagner, 2017) first revealed the fragility of neural networks to imperceptible input changes. PGD (Madry et al., 2018) formalized this under a saddle-point framework, becoming the standard for adversarial training. AutoAttack (Croce & Hein, 2020) further unified strong attacks, including PGD variants, into a reliable benchmark. In text tasks, adversarial methods must contend with discrete inputs. TextFooler (Jin et al., 2020) substitutes key words with semantically similar ones to mislead predictions, while CodeAttack (Jha & Reddy, 2023) adapts this idea to code-language models. More recently, PGD has been extended to LLMs via continuous relaxation (Geisler et al., 2024), enabling efficient attacks in embedding space. Beyond evaluation,

PGD has also been used as a regularizer to improve optimization. PTP (Chen et al., 2023a) applies PGD-style perturbations in the prompt embedding space to smooth training and enhance stability. Inspired by this, our work extends PGD to the multimodal embedding space, enabling unified gradient-based control over both visual and textual inputs.

# 3 CAUSAL ANALYSIS ON MODALITY INTERFERENCE

**Cross-Modality Competency Problems in Multimodal Large Language Models**  Competency problems describe scenarios where models rely on spurious correlations between isolated input features and output labels to make predictions, instead of leveraging meaningful interactions among multiple features (Gardner et al., 2021). We extend this concept to the multimodal setting by treating entire modalities (e.g., image $X_I$ or text $X_T$) as structured feature sources. We define the *Cross-Modality Competency* as an ability for MLLM to fairly evaluate and integrate all modalities, identifying which ones carry task-relevant signals while ignoring misleading or irrelevant ones. For instance, in a pure-text question answering task, the model receives both a question and an image, as is standard in MLLM input formats. However, the image is not required to answer the question. If the model relies on spurious visual cues—such as objects or scenes that frequently co-occur with certain answers—it violates the task's competency condition by grounding predictions in irrelevant modality signals. This manifests as *Modality Interference*, where the presence of an irrelevant but misleading modality disrupts the model's reasoning.

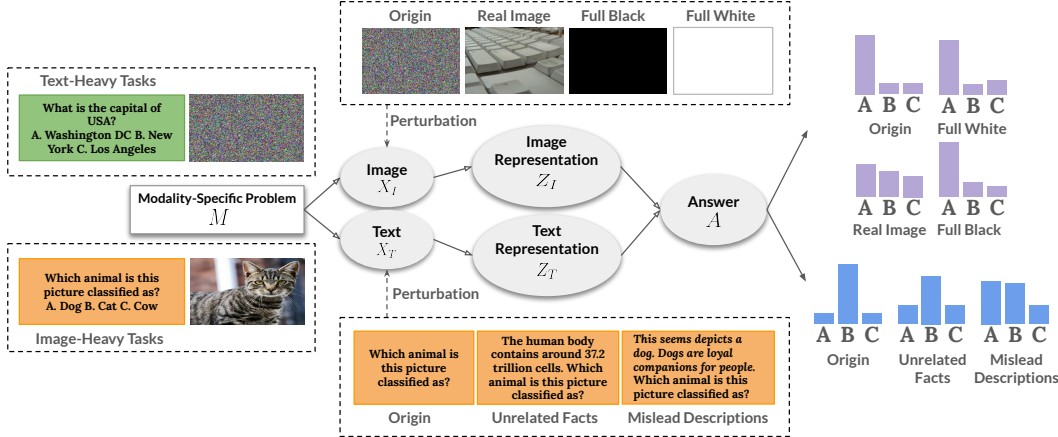

Figure 2: Causal graph illustrating modality interference in our perturbation-based evaluation analysis. Controlled interventions (heuristic) perturb either the image or text inputs, affecting their intermediate representations and ultimately the model prediction.

**Perturbation-based Evaluation Experiment**  To systematically measure cross-modality competency, we propose a perturbation-based evaluation framework. The core idea is to inject controlled noise into the irrelevant modality and assess the model's robustness to such perturbations. Specifically, for image-heavy tasks, we perturb the text input by: (1)Prepending *unrelated scientific facts*; (2) Prepending *misleading descriptions* that falsely link incorrect options to the image content. For text-heavy tasks, we perturb the visual input by: (1) Attaching a *real but irrelevant image*; (2) Substituting with a *full black* or *full white* canvas image. Models with strong modality selectivity should maintain high prediction consistency when irrelevant modality signals are perturbed. We select multiple image-heavy and text-heavy tasks for evaluation. Each task is framed as a multiple-choice classification problem, requiring the model to choose the correct option based on image and text modalities as input, with perturbations applied as described above. Details in §5, §C.1 and Tab. 8.

**Causal Modeling of Modality Interference**  We formalize modality interference through a causal intervention perspective with a causal graph, as shown in Fig. 2, where visual inputs ($X_I$) and textual inputs ($X_T$) are processed into their respective representations ($Z_I, Z_T$) before being fused to produce the final prediction ($A$). To study the model's reliance on different modalities, we introduce perturbations directly at the input level, serving as causal interventions (Pearl, 1995) on $X_I$ and $X_T$. Under ideal cross-modality competency, the model's prediction should primarily depend on the

task-relevant pathway (e.g., $X_I \rightarrow Z_I \rightarrow A$ in image-heavy tasks, $X_T \rightarrow Z_T \rightarrow A$ in text-heavy tasks). Causal interventions at the input level allow us to diagnose whether the model improperly fuses irrelevant signals into its decision process. We use $x'_I$ to denote an intervention on image $X_I$ and use $x'_T$ as an intervention on text $X_T$. Following Pearl's causal framework (Pearl, 1995; int, 2022), we quantify the impact of modality perturbations on model predictions by formalizing causal effects in our multimodal setting. Specifically, we define the pre-intervention prediction distribution as $P(A|X_I, X_T)$, and the post-intervention prediction distribution after applying a perturbation on $X_I$ or $X_T$ as $P'(A|\text{do}(X_I = x'_I) \text{ or } \text{do}(X_T = x'_T))$. The *do*-operation represents an intervention to specific modality, and the causal effect (CE) of an intervention is evaluated via a distance metric $\delta$ comparing $P$ and $P'$ as $\text{CE} = \delta(P, P')$. We assess the causal effect via prediction changes using $\delta_{\text{cp}}(P, P') := \mathbb{I}(a \neq a')$ in which $a = \arg\max_x P(x)$ is the predicted answer before intervention, $a' = \arg\max_x P'(x)$ is the predicted answer after intervention and $\mathbb{I}(\cdot)$ is the indicator function that outputs 1 if $a \neq a'$ and 0 otherwise. Thus, $\delta_{\text{cp}}$ captures whether the model's final decision $A$ changes under perturbations to the input modality. In all interventions, a high value of $\delta_{\text{cp}}$ indicates the model's susceptibility to modality interference, revealing spurious reliance on irrelevant modality.

## 4 METHODS

To mitigate modality interference and enhance cross-modality competency, we propose a unified perturbation-aware training framework that introduces interventions at both the input level (on $X_I$ and $X_T$) and the representation level (on $Z_I$ and $Z_T$) with consistency regularization. Overall pipeline is displayed in Fig. 3.

### 4.1 PERTURBATION-BASED DATA AUGMENTATION

To increase causal robustness along the desired paths $X_I \rightarrow Z_I \rightarrow A$ (for image-heavy tasks) or $X_T \rightarrow Z_T \rightarrow A$ (for text-heavy tasks), we first adopt a causally grounded data augmentation method by augmenting each sample with both heuristic perturbations and training time adversarial perturbations.

**Mixture of Multi-Task Training data with Heuristic Perturbations** Let $\mathcal{D} = (x_I, x_T, a)$ denote the full multimodal training dataset, where $(x_I, x_T)$ are the image and text inputs and $a$ is the ground-truth answer. The dataset $\mathcal{D}$ can be partitioned into

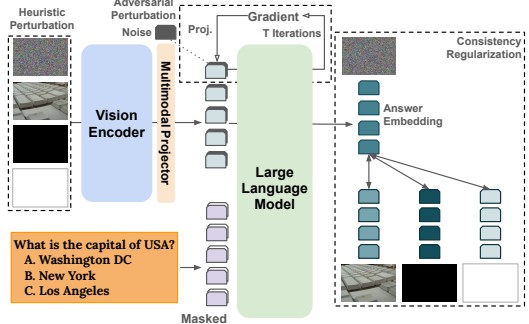

Figure 3: Overview of our proposed framework.

three subsets based on tasks: 1) Image-heavy set $\mathcal{D}^{\text{img}}$: samples where visual input $x_I$ is the dominant information source; 2) Text-heavy set $\mathcal{D}^{\text{text}}$: samples where textual input $x_T$ provides the main reasoning signal; 3) VQA set $\mathcal{D}^{\text{vqa}}$: samples from vision-language datasets with naturally balanced multimodal dependencies. In practice, we transform these image-heavy and text-heavy datasets into VQA format to construct $\mathcal{D}^{\text{img}}$ and $\mathcal{D}^{\text{text}}$, and derive VQA samples from the supervised finetuning stage of each MLLM as $\mathcal{D}^{\text{vqa}}$. For each sample $(x_I, x_T, a) \in \mathcal{D}^{\text{img}} \cup \mathcal{D}^{\text{text}}$, we maintain its original version and apply heuristic perturbation to construct Origin Samples and Perturbation-Augmented Samples. Origin samples are used to reinforce the desired causal path (e.g., $X_I \rightarrow Z_I \rightarrow A$). Perturbation-augmented samples are variants of the same instance with perturbations applied to the irrelevant modality, which are denoted as $(x_I, \tilde{x}_T, a) \in \mathcal{D}^{\text{img}}_{\text{pert}}$ and $(\tilde{x}_I, x_T, a) \in \mathcal{D}^{\text{text}}_{\text{pert}}$ where $\tilde{x}_T$ and $\tilde{x}_I$ are perturbed text and image respectively. Together, the augmented dataset can be written as:

$$\mathcal{D}^{\text{AUG}} = \mathcal{D}^{\text{img}} \cup \mathcal{D}^{\text{img}}_{\text{pert}} \cup \mathcal{D}^{\text{text}} \cup \mathcal{D}^{\text{text}}_{\text{pert}} \cup \mathcal{D}^{\text{VQA}}. \tag{1}$$

We sample $N_{\text{img}}$ and $N_{\text{text}}$ examples from $\mathcal{D}^{\text{img}}$ and $\mathcal{D}^{\text{text}}$ to construct $\mathcal{B}^{\text{img}}_{\text{orig}}$ and $\mathcal{B}^{\text{text}}_{\text{orig}}$ respectively, and the remaining $N_{\text{vqa}}$ examples are VQA samples from $\mathcal{D}^{\text{vqa}}$ to construct $\mathcal{B}^{\text{vqa}}$. With dynamically generated perturbed variants for each sample, the final training batch is:

$$\mathcal{B} = \mathcal{B}^{\text{img}}_{\text{orig}} \cup \mathcal{B}^{\text{img}}_{\text{pert}} \cup \mathcal{B}^{\text{text}}_{\text{orig}} \cup \mathcal{B}^{\text{text}}_{\text{pert}} \cup \mathcal{B}^{\text{vqa}}, \tag{2}$$

where $\mathcal{B}^{\text{img}}_{\text{pert}}$ and $\mathcal{B}^{\text{text}}_{\text{pert}}$ are perturbation-augmented variants generated from $\mathcal{B}^{\text{img}}_{\text{orig}}$ and $\mathcal{B}^{\text{text}}_{\text{orig}}$, respectively. For the full training batch $\mathcal{B}$, which includes both original and perturbed samples, we define

the supervised loss $\mathcal{L}_{\text{sft}}$ as the cross-entropy loss computed over all answer tokens in the ground-truth sequences. Let $\mathcal{L}_{\text{cls}}(x_I, x_T, a)$ denote the standard autoregressive loss for a sample $(x_I, x_T, a)$, then:

$$\mathcal{L}_{\text{sft}} = \frac{1}{|\mathcal{B}|} \sum_{(x_I, x_T, a) \in \mathcal{B}} \mathcal{L}_{\text{cls}}(x_I, x_T, a). \tag{3}$$

**Adversarial Perturbation with Cross-modality Masking**  While heuristic perturbations simulate realistic but limited modality noise at the input level, they may not fully capture the worst-case failure modes of MLLMs, especially under complex spurious alignments in the representation space. To overcome this limitation, we introduce a stronger and more generalizable intervention through adversarial training. These perturbations simulate worst-case alignment disruptions during training, serving as targeted interventions on latent nodes $(Z_I, Z_T)$ to reveal the *Direct Causal Effect* (DCE) of irrelevant modalities on $A$. By optimizing the model under such adversarial conditions, we reduce the model's reliance on spurious cross-modal signals and reinforce task-relevant causal pathways. Inspired by PGD (Madry et al., 2018; Chen et al., 2023b), we design a tailored perturbation strategy for multimodal token embeddings $(Z_I, Z_T)$. Unlike standard PGD that applies coarse sign-based updates, our method introduces two critical modifications: (1) *Modality-specific perturbation masking*, which restricts perturbations to task-irrelevant modalities via a binary mask, thereby transforming noise into targeted causal probes rather than indiscriminate corruption. (2) *Raw-gradient updates*, where we remove the sign operator and apply the raw gradient directly, yielding smoother, more diverse, and more realistic perturbations that better simulate modality interference. Formally, we construct perturbations $\delta = (\delta_I, \delta_T)$ in the latent space that maximize the model's predictive loss:

$$\delta = \underset{\|\delta\|_\infty \leq \epsilon}{\arg\max} \, \mathcal{L}_{\text{cls}}(f(Z_I + \delta_I, Z_T + \delta_T)), \tag{4}$$

where $\epsilon$ bounds the perturbation strength and $f$ is the prediction function. We optimize $\delta$ through $n$ raw-gradient steps, updating at each step $t$ as:

$$\delta^{(t+1)} = \Pi_{\|\delta\|_\infty \leq \epsilon} \left( \delta^{(t)} + \alpha \cdot \nabla_\delta \mathcal{L}_{\text{cls}}(f(Z + \delta^{(t)})) \right), \tag{5}$$

where $\alpha$ is the step size, $\Pi$ projects the noise into the $\ell_\infty$ ball, and $Z = [Z_I; Z_T]$ is the concatenated embedding. In practice, we integrate a modality-specific binary mask $M \in \{0, 1\}^{L \times d}$, where $L$ is the sequence length and $d$ the hidden dimension, ensuring that perturbations only affect task-irrelevant tokens. Given multimodal embeddings $\mathbf{E} \in \mathbb{R}^{L \times d}$, the perturbed embeddings are:

$$\tilde{\mathbf{E}} = \mathbf{E} + \delta \odot M, \tag{6}$$

with $\odot$ denoting element-wise masking. We initialize $\delta$ with Gaussian noise $\mathcal{N}(0, \epsilon^2)$ and update it for $T$ steps. The final adversarial objective is $\mathcal{L}_{\text{adv}} = \mathcal{L}_{\text{cls}}(f(\tilde{\mathbf{E}}))$.

## 4.2 Consistency Regularization under Perturbations

While perturbation-based augmentation exposes the model to diverse interventions, it does not constrain how intermediate representations $(Z_I, Z_T)$ should respond, and even small changes in the task-irrelevant modality may cause undesirable shifts in fused features. To address this, we introduce a consistency regularization strategy that enforces *output stability* between original and perturbed inputs, serving as an indirect constraint on $Z_I$ and $Z_T$ to mitigate modality interference. By minimizing the divergence between the prediction distributions of original and perturbed inputs, the model is encouraged to maintain invariant behavior along the task-relevant causal paths. Formally, given an original input $x$ and its perturbed counterpart $\tilde{x}$, with predictive distributions $p_\theta(A|x)$ and $p_\theta(A|\tilde{x})$, the consistency loss follows the general form:

$$\mathcal{L}_{\text{consistency}} = \text{Consistency}\big(p_\theta(A|x) \, \| \, p_\theta(A|\tilde{x})\big). \tag{7}$$

In practice, we instantiate this by applying distributional divergence (e.g., KL or JS) at the token level. Let $l^{\text{orig}}, l^{\text{pert}} \in \mathbb{R}^{L_A \times V}$ denote the pre-softmax logits for the original and perturbed samples, where $L_A$ is the number of answer tokens and $V$ the vocabulary size. Using KL divergence with temperature $\tau$ as an example, the loss becomes (equally apply to image-heavy and text-heavy tasks):

$$\mathcal{L}_{\text{consistency}} = \frac{1}{L_A} \sum_{i=1}^{L_A} \sum_{v=1}^{V} \text{softmax}\left(\frac{l_i^{\text{orig}}}{\tau}\right)_v \cdot \log \frac{\text{softmax}\left(\frac{l_i^{\text{orig}}}{\tau}\right)_v}{\text{softmax}\left(\frac{l_i^{\text{pert}}}{\tau}\right)_v}. \tag{8}$$

Table 1: Evaluation on unimodal and VQA datasets. For unimodal datasets, we report accuracy under the original input (Orig) and the worst-performing perturbation (Perturbed). For VQA datasets, we report accuracy on the original setting. Best results are highlighted in bold.

| Model Settings | Mini-ImageNet | | Caltech-101 | | OpenBookQA | | MMLU | | ScienceQA | MM-Bench | Seed-Bench |
|---|---|---|---|---|---|---|---|---|---|---|---|
| | Orig | Perturbed | Orig | Perturbed | Orig | Perturbed | Orig | Perturbed | Accuracy | Accuracy | Accuracy |
| LLaVA-1.5-7B | 95.3 | 43.5 | 97.0 | 57.4 | 62.4 | 56.4 | 46.3 | 45.2 | 64.5 | 64.3 | 63.4 |
| + CoT (Wei et al., 2022) | 81.7 | 28.9 | 80.9 | 36.0 | 38.8 | 38.9 | 39.6 | 38.7 | 64.7 | 65.2 | 64.1 |
| + VLMClassifier-1 (Zhang et al., 2024) | 15.1 | 0.0 | 15.6 | 0.0 | 61.5 | 61.5 | 47.8 | 47.5 | 61.1 | 36.2 | 35.9 |
| + VLMClassifier-2 (Zhang et al., 2024) | 15.6 | 0.0 | 15.1 | 0.0 | 61.8 | 61.2 | 47.4 | 47.8 | 61.8 | 35.8 | 36.0 |
| Ours | **98.6** | **98.4** | **99.3** | **98.9** | **81.8** | **81.0** | **51.5** | **51.0** | **67.8** | **73.1** | **64.6** |
| LLaVA-1.5-13B | 95.6 | 73.0 | 97.9 | 77.4 | 65.9 | 63.8 | 51.8 | 50.8 | 66.1 | 72.1 | 64.5 |
| + CoT (Wei et al., 2022) | 92.9 | 62.8 | 96.6 | 67.8 | 55.6 | 53.0 | 47.3 | 45.5 | 65.6 | 70.2 | 64.9 |
| + I-MoF (Tong et al., 2024) | 93.9 | 70.1 | 97.8 | 80.9 | 69.2 | 64.5 | 46.2 | 39.1 | **66.8** | 73.0 | 66.6 |
| Ours | **98.5** | **98.4** | **99.2** | **98.6** | **83.0** | **82.1** | **56.6** | **55.8** | 62.6 | **73.7** | **68.4** |

## 4.3 FINAL TRAINING OBJECTIVE

Our final training objective integrates both perturbation-based data augmentation and consistency regularization into a unified framework. For each batch, we begin with a set of original samples $\mathcal{B}_{\text{orig}}$ and dynamically construct their heuristic perturbed counterparts $\mathcal{B}_{\text{pert}}$ via input-level augmentations. We then apply adversarial perturbations on both $\mathcal{B}_{\text{orig}}$ and $\mathcal{B}_{\text{pert}}$, and enforce consistency between the predictions of original and all perturbed samples. The overall loss is:

$$\mathcal{L}_{\text{total}} = \mathcal{L}_{\text{sft}} + \mathcal{L}_{\text{adv}} + \lambda_{\text{cons}} \cdot \mathcal{L}_{\text{consistency}}, \tag{9}$$

where $\mathcal{L}_{\text{sft}}$ is the supervised loss computed over $\mathcal{B}$, $\mathcal{L}_{\text{adv}}$ is the adversarial loss computed on all adversarial perturbed samples and $\mathcal{L}_{\text{consistency}}$ is the consistency loss between original and all perturbed sample pairs. By aligning all three losses with the causal structure of multimodal reasoning, we systematically mitigate modality interference and improve cross-modality competency in MLLMs.

## 5 EXPERIMENTS

**Models.** We conduct experiments on three MLLM families with different parameter size: Qwen2.5-vl-3b (Bai et al., 2025), LLaVA-1.5-7B & LLaVA-1.5-13B (Liu et al., 2023a) and InstructBLIP-Vicuna-7B (Luo et al., 2023). Following LLaVA and Qwen-VL, we freeze the vision encoder and train the multimodal projector and language model; for InstructBLIP, we instead freeze both the vision encoder and Q-Former, fine-tuning only the projection layer and language model (see §B).

**Baselines.** We include following baselines for comparison: *LLaVA-1.5-13B + I-MoF* (Tong et al., 2024): By applying the designed Interleaved Mixture-of-Features (I-MoF) module on LLaVA-1.5-13B to spatially combine CLIP (Radford et al., 2021) and DINOv2 (Oquab et al., 2023) visual tokens, it enhances visual grounding by integrating complementary features from contrastive and self-supervised vision encoders. *VLMClassifier* (Zhang et al., 2024): it enhances visually-grounded language models for image classification by fine-tuning them on ImageNet (Deng et al., 2009) (*VLMClassifier-1*) or ImageNet combining LLaVA-Instruct (Liu et al., 2023b) (*VLMClassifier-2*). *Chain-of-Thought (CoT) Prompting* (Wei et al., 2022): we further evaluate prompt-based mitigation by encouraging structured reasoning through CoT-style prompting. The specific prompt design and results are reported in §C.4.

**Datasets.** We evaluate models on benchmarks covering three task types: **(i) Image-heavy tasks**: Mini-ImageNet (Russakovsky et al., 2015) and Caltech-101 (Fei-Fei et al., 2004), used in both training and evaluation, originally designed for image classification; **(ii) Text-heavy tasks**: OpenBookQA (Mihaylov et al., 2018) and MMLU (Hendrycks et al., 2020), consisting purely of textual question answering data; **(iii) VQA tasks**: For training, we use LLaVA-Instruct-dataset (Liu et al., 2023b) as the instruction-tuning dataset for related models. For InstructBLIP, we additionally use TextCaps (Sidorov et al., 2020) as another publicly available VQA dataset used in instruction tuning. For Qwen2.5-VL, whose instruction-tuning data is proprietary, we adopt LLaVA-Instruct as a standardized alternative. For evaluation, we adopt three multiple-choice VQA benchmarks: ScienceQA-IMG (Lu et al., 2022), MM-Bench-EN (Liu et al., 2023c), and Seed-Bench-IMG (Li et al., 2023). For ScienceQA and Seed-Bench, we only include examples with image context. For MM-Bench, we use the English version. All datasets are converted into a unified multiple-choice VQA format, enabling consistent modeling and evaluation across tasks and models. We report the accuracy of all multiple choice tasks and quantify the causal effect with the prediction change rate $\delta_{\text{cp}}$. All models are fine-tuned for 1 epoch with a fixed batch size $N_{\text{batch}}$. All the hyperparameters are listed in §D. For each dataset, results are averaged over multiple independent runs. Following standard practice, inference is performed with deterministic decoding (temperature fixed at 0).

Table 2: Multimodal reasoning accuracy (%) on VQA benchmarks under various ablation study configurations. The best accuracy is marked in bold. Arrows (↑ / ↓) indicate relative changes compared to the Vanilla baseline of each model. Overall performance is computed as a weighted average across datasets, with weights proportional to each dataset's test size Tab. 13.(full results in Tab. 7)

| Model | Method | ScienceQA-IMG | MM-Bench-EN | Seed-Bench-IMG | VQA Overall |
|---|---|---|---|---|---|
| *3B Multimodal Models* | | | | | |
| **Qwen2.5-VL-3B**
Bai et al. (2025) | Vanilla | 63.0 (100%) | 81.8 (100%) | 72.3 (100%) | 72.5 (100%) |
| | FFT with $D^{\text{VQA}}$ | 75.3 (↑19.5%) | 82.0 (↑0.2%) | 74.7 (↑3.3%) | 76.1 (↑5.0%) |
| | FFT with $D^{\text{AUG}}$ | 73.6 (↑16.8%) | 81.7 (↓0.1%) | 75.3 (↑4.1%) | 76.2 (↑5.1%) |
| | + KL | 72.8 (↑15.6%) | 80.9 (↓1.1%) | 74.9 (↑3.6%) | 75.9 (↑4.7%) |
| | + JS | 73.4 (↑16.5%) | 82.0 (↑0.2%) | 75.2 (↑4.0%) | 76.2 (↑5.1%) |
| | + RG | 73.4 (↑16.5%) | 81.8 (↑0.0%) | 75.0 (↑3.7%) | 76.0 (↑4.8%) |
| | + ADV | 73.1 (↑16.0%) | 81.7 (↓0.1%) | 75.3 (↑4.1%) | 76.1 (↑5.0%) |
| | Ours | 73.8 (↑17.1%) | 81.5 (↓0.4%) | 75.5 (↑4.4%) | **76.4 (↑5.4%)** |
| *7B Multimodal Models* | | | | | |
| **LLaVA-1.5-7B**
Liu et al. (2023a) | Vanilla | 64.5 (100%) | 64.3 (100%) | 63.4 (100%) | 63.8 (100%) |
| | FFT with $D^{\text{VQA}}$ | 61.6 (↓4.5%) | 71.4 (↑11.0%) | 62.4 (↓1.6%) | 64.0 (↑0.3%) |
| | FFT with $D^{\text{AUG}}$ | 65.4 (↑1.4%) | 71.1 (↑10.6%) | 63.9 (↑0.8%) | 65.6 (↑2.8%) |
| | + KL | 65.9 (↑2.2%) | 72.5 (↑12.8%) | 64.5 (↑1.7%) | 66.3 (↑3.9%) |
| | + JS | 63.8 (↓1.1%) | 73.5 (↑14.3%) | 63.7 (↑0.5%) | 65.6 (↑2.8%) |
| | + RG | 66.3 (↑2.8%) | 71.8 (↑11.7%) | 63.7 (↑0.5%) | 65.7 (↑3.0%) |
| | + ADV | 66.7 (↑3.4%) | 71.4 (↑11.0%) | 63.6 (↑0.3%) | 65.7 (↑3.0%) |
| | Ours | 67.8 (↑5.1%) | 73.1 (↑13.7%) | 64.6 (↑1.9%) | **66.8 (↑4.7%)** |
| **InstructBlip-7B**
Luo et al. (2023) | Vanilla | 52.1 (100%) | 65.5 (100%) | 54.8 (100%) | 56.4 (100%) |
| | FFT with $D^{\text{VQA}}$ | 61.8 (↑18.6%) | 68.4 (↑4.4%) | 57.6 (↑5.1%) | 60.4 (↑7.1%) |
| | FFT with $D^{\text{AUG}}$ | 65.4 (↑25.5%) | 71.1 (↑8.5%) | 61.2 (↑11.7%) | 63.9 (↑13.3%) |
| | + KL | 65.9 (↑26.5%) | 71.7 (↑9.5%) | 60.9 (↑11.1%) | 63.9 (↑13.3%) |
| | + JS | 66.9 (↑28.4%) | 72.3 (↑10.4%) | 60.7 (↑10.8%) | 64.1 (↑13.7%) |
| | + RG | 62.8 (↑20.5%) | 70.9 (↑8.2%) | 62.5 (↑14.0%) | 64.2 (↑13.8%) |
| | + ADV | 66.0 (↑26.7%) | 69.0 (↑5.3%) | 61.8 (↑12.8%) | 64.0 (↑13.5%) |
| | Ours | 64.0 (↑22.8%) | 71.4 (↑9.0%) | 63.0 (↑14.9%) | **64.8 (↑14.9%)** |
| *13B Multimodal Models* | | | | | |
| **LLaVA-1.5-13B**
Liu et al. (2023a) | Vanilla | 65.8 (100%) | 72.1 (100%) | 64.5 (100%) | 66.2 (100%) |
| | FFT with $D^{\text{VQA}}$ | 60.8 (↓7.6%) | 73.6 (↑2.1%) | 64.9 (↑0.6%) | 65.8 (↓0.6%) |
| | FFT with $D^{\text{AUG}}$ | 63.5 (↓3.5%) | 75.0 (↑4.0%) | 65.7 (↑1.9%) | 67.1 (↑1.4%) |
| | + KL | 62.5 (↓5.0%) | 74.6 (↑3.5%) | 67.6 (↑4.8%) | 68.0 (↑2.7%) |
| | + JS | 65.8 (↑0.0%) | 74.7 (↑3.6%) | 67.6 (↑4.8%) | **68.6 (↑3.6%)** |
| | + RG | 58.3 (↓11.4%) | 73.5 (↑1.9%) | 67.4 (↑4.5%) | 66.9 (↑1.1%) |
| | + ADV | 57.6 (↓12.5%) | 74.2 (↑2.9%) | 68.3 (↑5.9%) | 67.5 (↑2.0%) |
| | Ours | 62.6 (↓4.9%) | 73.7 (↑2.2%) | 68.4 (↑6.0%) | 68.4 (↑3.3%) |

**Achieving Pareto-Optimality Across Unimodal and Multimodal Tasks**  As shown in Tab. 1, our method outperforms all baselines across different base MLLMs, demonstrating stronger robustness to modality interference and improved cross-modality competency. While CoT slightly improves performance in certain VQA settings, its overall gains are minimal and inconsistent, and it fails to mitigate modality interference under perturbed conditions (e.g., 85.9% vs. 97.9% on Mini-ImageNet). While I-MoF enhances visual grounding by integrating multiple visual features, it still suffers from modality interference: e.g. LLaVA-1.5-13B + I-MoF achieves 93.9% on original Mini-ImageNet but drops to 70.1% under perturbation (↓23.8%), indicating reliance on spurious textual cues. In contrast, our method maintains perturbed performance at 98.4% (↓0.1%). On the other hand, VLMClassifier, adopts vision-only fine-tuning, which leads to two critical limitations: vulnerability to cross-modal interference and degradation on VQA tasks, as LLaVA-1.5-7b + VLMClassifier-1 only reaches 35.8%/36.2% on MM-Bench/SeedBench, notably lower than both base LLaVA and our method (73.7%/68.4%). These results highlight that vision-centric strategies, without addressing modality alignment, are insufficient for robust multimodal understanding. In text-heavy tasks such as OpenBookQA and MMLU, our method also achieves superior perturbed performance(e.g., 55.8% vs. 39.1% on MMLU on LLaVA-1.5-13B)—highlighting that addressing modality interference directly, rather than merely improving representations, is key to robust multimodal reasoning. Overall, unlike prior methods that often trade off between unimodal and multimodal performance, our method consistently improves both, achieving Pareto-optimality.

**Ablation Studies**  To evaluate the effectiveness of each component in our framework, we conduct a comprehensive ablation study across multiple models and scales. We compare the pretrained models with the following strategies: *FFT with $D^{VQA}$* (standard finetuning on VQA data), *FFT with $D^{AUG}$* (supervised finetuning on mixed multi-task datasets with heuristic perturbations), *FFT+KL/JS*

Table 3: Evaluation of Caltech-101 (image-heavy) and MMLU (text-heavy) across different ablation study settings (full results in Tab. 6).

| Model | Method | Caltech-101 | | MMLU | |
|---|---|---|---|---|---|
| | | Orig | Perturbed | Orig | Perturbed |
| LLaVA-1.5-7B | Vanilla | 97.0 | 57.4 | 46.3 | 45.2 |
| | +$D^{VQA}$ | 96.2 | 46.3 | 46.8 | 45.6 |
| | +$D^{AUG}$ | 98.5 | 98.6 | 51.1 | 50.7 |
| | +KL | 98.6 | 98.7 | 51.1 | 50.7 |
| | +ADV | 98.7 | 98.5 | 50.6 | 50.3 |
| | Ours | **99.3** | **99.0** | **51.5** | 51.0 |
| InstructBLIP-7B | Vanilla | 90.3 | 17.5 | 35.3 | 35.2 |
| | +$D^{VQA}$ | 92.1 | 23.1 | 40.9 | 40.2 |
| | +$D^{AUG}$ | 99.0 | 56.1 | 50.0 | 49.7 |
| | +JS | 98.9 | **98.4** | **50.7** | 49.3 |
| | +ADV | 99.1 | 85.2 | 49.3 | 48.4 |
| | Ours | **99.2** | 98.3 | 50.2 | **49.7** |

Table 4: OOD robustness evaluation on Caltech-101 and MMLU, with OCR noise and Screenshot distractors (full results in Tab. 11).

| Model | Method | Caltech-101 | | MMLU | |
|---|---|---|---|---|---|
| | | Orig | OCR | Orig | Screenshot |
| LLaVA-1.5-7B | Vanilla | 97.0 | 92.8 | 46.3 | 44.8 |
| | +$D^{AUG}$ | 98.5 | 98.0 | 51.1 | 50.6 |
| | +ADV | 98.7 | 98.4 | 50.6 | 51.3 |
| | Ours | **99.3** | **99.0** | **51.5** | 51.3 |
| InstructBLIP-7B | Vanilla | 90.3 | 83.6 | 35.3 | 34.9 |
| | +$D^{AUG}$ | 99.0 | 98.4 | 50.0 | 49.2 |
| | +ADV | 99.1 | 98.8 | 49.3 | **49.3** |
| | Ours | **99.2** | **99.0** | **50.2** | 49.2 |
| Qwen2.5-VL-7B | Vanilla | 99.1 | 99.2 | 69.3 | 57.0 |
| | +$D^{AUG}$ | **99.7** | 99.3 | 70.4 | 63.7 |
| | +ADV | 99.6 | **99.6** | **70.4** | 69.7 |
| | Ours | 99.6 | 99.5 | 69.8 | **69.7** |

(adding consistency regularization on KL or JS divergence), *FFT+RG* (injecting random Gaussian noise into token embeddings), *FFT+ADV* (FFT with heuristic & adversarial perturbations), and *Ours* (combining both perturbation-based data augmentation and consistency regularization). Tab. 2 presents overall VQA performance, and Tab. 3 evaluates model robustness under unimodal settings. ( Tab. 5 reports results with all perturbations.) Together, these results show the effectiveness of our method in improving both general VQA accuracy and robustness under modality interference. Across all model families (Qwen2.5-VL, InstructBLIP, LLaVA-1.5) and model sizes (3B/7B/13B), our method consistently achieves best overall performance, improving accuracy on both unimodal and multimodal benchmarks. For instance, it boosts overall VQA accuracy (e.g., +14.9% on InstructBLIP-7B), but also enhances robustness to modality interference–improve the performance under perturbations by over 50% on image-heavy tasks(e.g. 17.5% → 98.3% with InstructBLIP-7B on Caltech101). We also extend evaluation from MCQA to free-form QA (§C.5).

We observe consistent improvements across both unimodal and multimodal tasks when moving from *FFT w/ $D^{VQA}$* to *FFT w/ $D^{AUG}$*, highlighting the importance of incorporating modality-specific supervision and heuristic perturbations. Building upon this, adding consistency regularization yields further gains by stabilizing model predictions under controlled perturbations on $X_I$ or $X_T$. Both KL and JS objectives lead to similar improvements, suggesting that the model equally benefits from all heuristic perturbations regardless of anchor choice.Finally, we compare adversarial perturbations with random Gaussian noise, and find that *FFT+ADV* consistently outperforms *FFT+RG* across most backbones, indicating that structured perturbations more effectively suppress spurious shortcuts and promote robust, task-relevant representations. To further validate generalization, we introduce two real-world out-of-distribution perturbations at test time: (i) noisy OCR snippets sampled from FUNSD (Jaume et al., 2019) as irrelevant text into image-heavy tasks; and (ii) unrelated UI screenshots from RICO (Deka et al., 2017) as distractor images in text-heavy tasks. As shown in Tab. 4, adversarial training significantly improves robustness under these unseen perturbations with consistent gains (e.g., on InstructBLIP-7B, 83.6% → 99.0% under OCR noise). These results demonstrate that **the modest overhead of adversarial training** (see §E) **yields substantial gains in out-of-domain generalization**, a crucial property for reliable deployment.

# 6 CONCLUSION

In this paper, we identify and formalize modality interference as a concrete manifestation of the broader cross-modality competency problem in Multimodal Large Language Models—namely, the inability to distinguish task-relevant from irrelevant modality signals. Through a designed perturbation-based causal evaluation experiment, we demonstrate that even state-of-the-art MLLMs systematically exhibit degraded performance under irrelevant but misleading inputs, revealing a fundamental vulnerability in their inference-time reasoning. To mitigate this issue, we propose a robust fine-tuning strategy that combines modality-specific data augmentation, consistency regularization, and adversarial perturbation in the embedding space. These designs explicitly constrain the model to produce stable outputs under spurious modality shifts, thereby reducing reliance on non-causal correlations and improving robustness. Extensive experiments across diverse architectures, scales, and task regimes confirm that our approach consistently improves both unimodal reasoning and multimodal generalization, achieving Pareto-optimal performance.

## ETHICS STATEMENT

This work adheres to the ICLR Code of Ethics.[1] Our research focuses on analyzing and mitigating modality interference in Multimodal Large Language Models. All experiments are conducted on publicly available benchmark datasets, including Mini-ImageNet, Caltech-101, OpenBookQA, MMLU, ScienceQA, MM-Bench, and Seed-Bench, which contain no personally identifiable or sensitive information beyond what is publicly released. We do not foresee direct risks of harm to individuals or groups arising from this research. Nevertheless, potential societal impacts include bias amplification or misinterpretation when deploying MLLMs in real-world applications. We note these risks and emphasize that our contributions are methodological and diagnostic rather than application-specific. No human subjects were involved, and no IRB approval was required. All funding sources are acknowledged in the main paper. Further discussions of limitations, broader impacts and LLM use are provided in Appendix §F, Appendix §G and Appendix §H.

## REPRODUCIBILITY STATEMENT

We are committed to ensuring the reproducibility of our work.

- **Code and Implementation:** We will release a full open-source codebase, including data processing, training, and evaluation scripts, upon publication.
- **Datasets:** All datasets used in this work are publicly available (Mini-ImageNet, Caltech-101, OpenBookQA, MMLU, ScienceQA, MM-Bench, Seed-Bench). Detailed preprocessing steps and dataset conversions into unified multiple-choice VQA format are described in §5 and Appendix §C.
- **Model and Training Details:** Hyperparameters (learning rates, batch sizes, epochs, optimizer choices) and architectural specifications are reported in §5 and Appendix §B, §D, §E.
- **Evaluation:** Metrics, baselines, and evaluation protocols are fully documented in §5 and Appendix §C with complete ablation results.

Together, these materials should enable independent researchers to reproduce our findings.

---

[1]https://iclr.cc/public/CodeOfEthics

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

# A APPENDIX SUMMARY

This appendix provides comprehensive supplementary materials and discussion to support the main findings of our paper on diagnosing and mitigating modality interference in MLLMs. We organize the appendix into several sections:

**Finetuning Strategies** (§B): We elaborate on our design choice to freeze the Q-Former in InstructBLIP-based models. This decision is motivated by the need to retain strong visual representations while avoiding overfitting to perturbed or misleading multimodal inputs. ( Tab. 13 records the size for each dataset)

**Detailed Experimental Results** (§C): This section includes three key tables— Tab. 5, Tab. 6 and Tab. 7—which report model performance on unimodal and multimodal tasks under various perturbation settings and ablation conditions(additional models included). Tab. 8 records the performance of different vanilla MLLMs under modality interference across modality-heavy datasets. We also include radar plots ( Fig. 4) that visualize task-wise robustness across different MLLMs. We provided the detailed experimental results on Qwen2.5-VL-7b (Bai et al., 2025) and InstructBlip-Vicuna-13b (Luo et al., 2023) and make further discussion on the selection of specific consistency loss. In Tab. 11, we examine the generalization benefits of adversarial training by evaluating robustness under two types of out-of-distribution (OOD) perturbations: real-world OCR noise (from FUNSD (Jaume et al., 2019)) and unrelated screenshots (from RICO (Deka et al., 2017)). In Tab. 9, we assess the impact of Chain-of-Thought prompting in mitigating modality interference, comparing its effectiveness against our method and standard baselines across both visual and textual modalities. In Tab. 10, we report results on the free-form generative VQA benchmark TextVQA (Singh et al., 2019), highlighting our method's generalizability beyond multiple-choice formats.

**Hyperparameter Settings** (§D): We present full training configurations used in our experiments, including optimization strategies, perturbation settings, and sampling ratios for different task types. This section enables reproducibility and highlights the computational efficiency of our proposed training scheme. We provide parameter analysis on iterations of adversarial training in Fig. 5.

**Compute Resource Details** (§E): We document hardware specifications, training durations, and resource costs for models of different scales. These details contextualize the feasibility of our approach in academic environments.

**Limitations** (§F): We discuss the granularity of our current modality interference analysis, the selections of perturbations, and propose directions for more fine-grained future studies.

**Broader Impacts** (§G): We reflect on the ethical implications and societal benefits of our research. While our methods improve model robustness and alignment, we also acknowledge the dual-use nature of adversarial perturbations and advocate for safety-aware deployment.

**LLM Use** (§H): Finally, we clarify that LLMs were only used to polish the writing of this paper.

Together, these sections provide a complete view of our technical contributions, empirical findings, and responsible research considerations.

# B FINETUNING STRATEGIES

In our adaptation of InstructBLIP-Vicuna-7B, we choose to freeze the Q-Former and only fine-tune the language model and the projection layer. This decision is grounded in the nature of the Q-Former as a highly task-specific visual query encoder, originally pre-trained on VQA-style datasets where fine-grained and semantically aligned image-text pairs dominate.

However, in our setting, we deliberately introduce perturbations to the input modalities (e.g., injecting unrelated or misleading text/image content), which breaks the expected alignment structure. We observe that training the Q-Former under such noisy supervision leads to unstable representations and overfitting to spurious modality correlations. In contrast, freezing the Q-Former allows us to preserve its original strong visual grounding capabilities, while letting the downstream language model learn to filter or suppress misleading signals introduced during training.

This alternative tuning strategy enhances robustness under modality interference and aligns with our overall goal of improving cross-modal competency in MLLMs under perturbed conditions.

Table 5: Unimodal ability evaluation on image-heavy and text-heavy tasks under perturbation. Left: Mini-ImageNet and Caltech-101; Right: OpenBookQA and MMLU. UF = Unrelated Facts, MD = Misleading Descriptions, RP = Random Pixels, RI = Real Image, FB & FW = Full Black/White Canvas. The best accuracy is marked in bold.

| Model | Method | Mini-ImageNet | | | Caltech-101 | | | OpenBookQA | | | | MMLU | | | |
|---|---|---|---|---|---|---|---|---|---|---|---|---|---|---|---|
| | | Orig | UF | MD | Orig | UF | MD | RP | RI | FB | FW | RP | RI | FB | FW |
| **Qwen2.5-VL-3B** Bai et al. (2025) | Vanilla | 98.9 | 98.5 | 94.9 | 98.8 | 99.0 | 94.4 | 79.9 | 74.6 | 80.0 | 79.7 | 63.5 | 61.1 | 64.0 | 63.6 |
| | FFT with $D^{\text{VQA}}$ | 98.8 | 98.5 | 95.3 | 98.8 | 98.1 | 94.3 | 80.7 | 74.3 | 80.1 | 80.2 | 63.0 | 61.7 | 63.0 | 63.3 |
| | FFT with $D^{\text{AUG}}$ | 98.8 | 98.8 | 98.6 | 99.6 | 99.3 | 99.6 | **87.1** | 86.7 | **87.4** | **87.2** | 64.8 | 63.9 | 64.8 | 64.7 |
| | + KL | 98.9 | 98.7 | 99.1 | 99.6 | 99.5 | **99.7** | 87.1 | 86.2 | 86.7 | 86.8 | **66.0** | **65.5** | **65.9** | **65.9** |
| | + JS | 99.1 | 98.8 | 98.3 | 99.6 | 99.4 | 98.0 | 85.0 | 84.2 | 85.1 | 85.1 | 65.6 | 65.1 | 65.5 | 65.5 |
| | + RG ($\sigma$=0.05) | 99.0 | 99.1 | 98.9 | 99.5 | 99.5 | 99.2 | 86.4 | **86.9** | 86.9 | **87.2** | 64.6 | 64.3 | 64.6 | 64.8 |
| | + ADV | **99.3** | **99.3** | 99.1 | 99.5 | 99.4 | 99.5 | 86.6 | 85.8 | 86.8 | 86.6 | 65.3 | 64.0 | 65.4 | 65.3 |
| | Ours | **99.3** | 99.2 | **99.2** | **99.7** | **99.7** | 99.5 | 86.7 | 86.4 | 86.6 | 86.6 | 64.8 | 64.5 | 65.0 | 65.1 |
| **Qwen2.5-VL-7B** Bai et al. (2025) | Vanilla | 99.3 | 99.3 | 96.3 | 99.1 | 98.9 | 97.2 | 85.9 | 77.5 | 85.8 | 86.0 | 69.3 | 63.7 | 68.9 | 68.9 |
| | FFT with $D^{\text{VQA}}$ | 99.2 | 99.3 | 96.0 | 99.5 | 99.5 | 95.7 | 86.3 | 82.3 | 86.5 | 86.3 | 69.2 | 67.4 | 69.4 | 69.3 |
| | FFT with $D^{\text{AUG}}$ | **99.6** | **99.5** | 99.4 | **99.7** | **99.7** | 99.5 | 90.2 | 90.2 | 90.3 | 90.3 | 70.4 | 69.9 | 70.4 | 70.3 |
| | + KL | 99.3 | 99.3 | 99.1 | 99.6 | 99.6 | 99.6 | **92.0** | 91.7 | **92.2** | **92.1** | 71.2 | **70.7** | 71.0 | 71.0 |
| | + JS | 99.5 | 99.4 | 99.4 | **99.7** | 99.3 | 99.6 | 91.6 | 92.1 | **92.2** | **92.1** | **71.5** | 69.9 | **71.5** | **71.6** |
| | + RG ($\sigma$=0.05) | 99.4 | 99.4 | 99.2 | 99.6 | 99.3 | 99.4 | 89.1 | 87.9 | 89.1 | 89.1 | 66.7 | 65.6 | 66.5 | 66.5 |
| | + ADV | 99.4 | 99.3 | 99.2 | 99.6 | 99.5 | 99.6 | 91.7 | **92.2** | 91.8 | 91.8 | 70.4 | 70.0 | 70.4 | 70.4 |
| | Ours | **99.6** | **99.5** | **99.5** | 99.6 | 99.6 | **99.7** | 90.9 | 89.7 | 90.8 | 91.0 | 69.8 | 68.2 | 69.8 | 70.0 |
| **LLaVA-1.5-7B** Liu et al. (2023a) | Vanilla | 95.3 | 93.4 | 43.5 | 97.0 | 95.9 | 57.4 | 62.4 | 56.4 | 62.5 | 63.4 | 46.3 | 45.2 | 45.9 | 45.8 |
| | FFT with $D^{\text{VQA}}$ | 94.3 | 92.7 | 41.5 | 96.2 | 94.0 | 46.3 | 61.3 | 55.5 | 62.0 | 62.9 | 46.8 | 45.6 | 47.5 | 47.7 |
| | FFT with $D^{\text{AUG}}$ | 98.2 | 98.2 | 98.1 | 98.5 | 98.6 | 99.0 | 78.6 | 77.2 | 78.7 | 78.4 | 51.1 | 50.7 | 51.1 | 51.3 |
| | + KL | **99.1** | **99.0** | 98.9 | 98.6 | 98.7 | 98.8 | 81.4 | 81.3 | 81.4 | 81.2 | **52.0** | 51.8 | 52.0 | 52.2 |
| | + JS | 98.7 | 98.8 | **99.0** | 99.1 | **99.0** | 99.2 | 81.6 | **81.6** | 81.7 | 81.5 | 51.6 | **51.8** | **52.4** | **52.4** |
| | + RG ($\sigma$=0.05) | 98.4 | 98.4 | 98.5 | 98.9 | 98.9 | 99.1 | 80.5 | 79.8 | 79.9 | 80.3 | 49.5 | 49.5 | 49.9 | 49.6 |
| | + ADV | 98.7 | 98.7 | 98.5 | 98.9 | 98.9 | 98.8 | 81.7 | 81.0 | 81.4 | 80.8 | 50.6 | 50.3 | 50.9 | 50.7 |
| | Ours | 98.6 | 98.4 | 98.7 | **99.3** | 98.9 | **99.3** | **81.8** | 81.0 | **81.7** | **81.7** | 51.5 | 50.9 | 51.5 | 51.4 |
| **LLaVA-1.5-13B** Liu et al. (2023a) | Vanilla | 95.6 | 94.1 | 73.0 | 97.9 | 97.1 | 77.4 | 65.9 | 63.8 | 68.0 | 69.1 | 51.8 | 50.8 | 52.7 | 52.7 |
| | FFT with $D^{\text{VQA}}$ | 94.6 | 93.9 | 72.0 | 97.8 | 96.5 | 80.2 | 67.5 | 64.2 | 69.1 | 69.3 | 52.4 | 52.2 | 53.1 | 53.3 |
| | FFT with $D^{\text{AUG}}$ | 98.1 | 96.8 | 98.4 | 96.7 | 96.9 | 97.0 | 81.0 | 78.7 | 81.1 | 81.3 | 52.1 | 51.7 | 51.8 | 51.6 |
| | + KL | 98.3 | 98.0 | 98.6 | 98.8 | 98.5 | 98.9 | 83.0 | **82.6** | **83.3** | 83.0 | 55.7 | 55.1 | 55.6 | 55.6 |
| | + JS | 98.3 | 98.1 | 98.0 | 98.7 | 98.4 | 98.7 | **83.1** | 81.5 | 83.1 | **83.1** | **56.7** | **56.2** | 56.6 | 56.5 |
| | + RG ($\sigma$=0.05) | 98.5 | 98.0s | 98.1 | 98.9 | 98.5 | 98.9 | **83.5** | 82.5 | 83.1 | 82.8 | 55.4 | 55.3 | 55.7 | 55.5 |
| | + ADV | **98.7** | 98.2 | 98.6 | 99.0 | **98.6** | 99.0 | 82.2 | **82.6** | 82.6 | 82.8 | 55.6 | 55.4 | 55.6 | 55.5 |
| | Ours | 98.5 | **98.4** | **98.7** | **99.2** | **98.6** | **99.2** | 83.0 | 82.1 | 82.7 | **83.1** | **56.7** | 55.8 | **56.7** | **56.7** |
| **InstructBlip-7B** Luo et al. (2023) | Vinilla | 92.0 | 87.1 | 13.6 | 90.3 | 90.2 | 17.5 | 50.8 | 46.2 | 50.9 | 50.7 | 35.3 | 35.8 | 35.2 | 35.7 |
| | FFT with $D^{\text{VQA}}$ | 95.6 | 86.6 | 16.3 | 98.3 | 91.0 | 23.1 | 49.8 | 45.2 | 49.5 | 50.7 | 40.9 | 40.2 | 41.0 | 41.6 |
| | FFT with $D^{\text{AUG}}$ | 98.5 | 98.0 | 38.2 | 99.0 | 98.7 | 56.1 | 75.0 | 74.9 | 74.8 | 75.8 | 50.0 | 49.7 | 50.0 | 50.0 |
| | + KL | 98.7 | 98.1 | **98.3** | **99.5** | **99.0** | **99.6** | 76.9 | 77.0 | 76.9 | 77.3 | **51.3** | **50.6** | **51.3** | **51.5** |
| | + JS | 98.5 | 97.7 | **98.5** | 98.9 | 98.4 | 98.9 | 78.0 | 76.6 | 77.7 | 78.0 | 50.7 | 50.1 | 50.7 | 50.8 |
| | + RG ($\sigma$=0.05) | **98.9** | 97.2 | 72.5 | 99.1 | 99.0 | 82.2 | 75.2 | 72.6 | 76.0 | 76.9 | 48.3 | 47.6 | 48.9 | 49.1 |
| | + ADV | 98.7 | **98.5** | 32.2 | **99.5** | 98.9 | 49.2 | 76.8 | 76.8 | 76.5 | 76.3 | 49.3 | 48.4 | 49.5 | 49.4 |
| | Ours | 98.4 | 97.9 | 98.0 | 99.2 | 98.3 | 99.0 | **79.0** | **77.3** | **79.3** | **79.0** | 50.2 | 49.7 | 50.3 | 50.2 |
| **InstructBlip-13B** Luo et al. (2023) | Vanilla | 95.6 | 94.1 | 73.0 | 97.9 | 97.1 | 77.4 | 65.9 | 63.8 | 68.0 | 69.1 | 51.8 | 50.8 | 52.7 | 52.7 |
| | FFT with $D^{\text{VQA}}$ | 95.6 | 85.8 | 8.0 | 97.0 | 87.5 | 11.6 | 58.6 | 55.4 | 59.7 | 60.6 | 43.7 | 42.8 | 43.6 | 44.1 |
| | FFT with $D^{\text{AUG}}$ | 98.4 | 98.2 | 9.3 | 99.2 | 98.8 | 13.8 | 82.0 | 80.4 | 81.2 | 81.2 | 52.1 | 51.3 | 52.4 | 53.0 |
| | + KL | 98.5 | **98.3** | 98.4 | 99.1 | 99.2 | **99.5** | 82.5 | 81.4 | 82.1 | 82.9 | **53.4** | **52.5** | **53.4** | 53.4 |
| | + JS | **98.7** | 98.1 | **98.9** | 99.3 | 99.2 | **99.5** | **83.5** | **83.1** | 83.1 | **83.3** | 52.8 | 52.2 | 53.2 | 53.3 |
| | + RG ($\sigma$=0.05) | 98.4 | 97.8 | 87.0 | **99.4** | **99.3** | 94.4 | 80.0 | 76.6 | 79.6 | 80.4 | 50.9 | 50.0 | 51.4 | 51.8 |
| | + ADV | 98.6 | 98.0 | 80.9 | 98.7 | 98.6 | 99.1 | 79.8 | 79.0 | 80.9 | 80.9 | 51.3 | 50.7 | 51.4 | 52.4 |
| | Ours | **98.7** | 97.9 | 98.0 | 98.7 | 98.7 | 98.8 | 83.2 | 81.2 | **83.8** | 83.0 | 52.2 | 51.6 | 52.3 | **53.4** |

# C DETAILED EXPERIMENTAL RESULTS

Please see  Tab. 5,  Tab. 6,  Fig. 4,  Tab. 8,  Tab. 7,  Tab. 11,  Tab. 9 and  Tab. 10 for more details.

Table 6: Evaluation of unimodal and multimodal tasks across different ablation study settings. For unimodal datasets, we report accuracy on the original setting (Orig) and the worst-performing perturbation (Perturbed). For VQA datasets, we report accuracy on the original setting. The best accuracy is marked in bold.

| Model | Method | Mini-ImageNet | | Caltech-101 | | OpenBookQA | | MMLU | | VQA Overall |
| | | Orig | Perturbed | Orig | Perturbed | Orig | Perturbed | Orig | Perturbed | Accuracy |
|---|---|---|---|---|---|---|---|---|---|---|
| *3B Multimodal Models* | | | | | | | | | | |
| **Qwen2.5-VL-3B** 
 Bai et al. (2025) | Vanilla | 98.9 | 94.9 | 98.8 | 94.4 | 79.9 | 74.6 | 63.5 | 61.1 | 72.5 |
| | FFT with $D^{VQA}$ | 98.8 | 95.3 | 98.8 | 94.3 | 80.7 | 74.3 | 63.0 | 61.7 | 76.1 |
| | FFT with $D^{AUG}$ | 98.8 | 98.6 | 99.6 | 99.3 | 87.1 | 86.2 | 64.8 | 63.9 | 76.2 |
| | + KL | 98.9 | 98.7 | 99.6 | 99.5 | **87.1** | **86.7** | **66.0** | **65.5** | 75.9 |
| | + JS | 99.1 | 98.3 | 99.6 | 98.0 | 85.0 | 84.2 | 65.6 | 65.1 | 76.2 |
| | + RG | 99.0 | 98.9 | 99.5 | 99.2 | 86.4 | 86.4 | 64.6 | 64.3 | 76.0 |
| | + ADV | 99.3 | 99.1 | 99.5 | 99.4 | 86.6 | 85.8 | 65.3 | 64 | 76.1 |
| | Ours | **99.3** | **99.2** | **99.7** | **99.5** | 86.7 | 86.6 | 64.8 | 64.5 | **76.4** |
| *7B Multimodal Models* | | | | | | | | | | |
| **LLaVA-1.5-7B** 
 Liu et al. (2023a) | Vanilla | 95.3 | 43.5 | 97.0 | 57.4 | 62.4 | 56.4 | 46.3 | 45.2 | 63.8 |
| | FFT with $D^{VQA}$ | 94.3 | 41.5 | 96.2 | 46.3 | 61.3 | 55.5 | 46.8 | 45.6 | 64.0 |
| | FFT with $D^{AUG}$ | 98.2 | 98.1 | 98.5 | 98.6 | 78.6 | 77.2 | 51.1 | 50.7 | 65.6 |
| | + KL | **99.1** | **99.0** | 98.6 | 98.7 | 81.4 | 81.2 | 51.1 | 50.7 | 66.3 |
| | + JS | 98.7 | 98.8 | 99.1 | 99.0 | 81.6 | 81.5 | **52.0** | **51.8** | 65.6 |
| | + RG | 98.4 | 98.4 | 98.9 | 98.9 | 80.5 | 79.8 | 49.5 | 49.5 | 65.7 |
| | + ADV | 98.7 | 98.5 | 98.7 | 98.5 | 81.7 | 80.8 | 50.6 | 50.3 | 65.7 |
| | Ours | 98.6 | 98.6 | **99.3** | **99.0** | **81.8** | **81.5** | 51.5 | 51.0 | **66.8** |
| **InstructBlip-7B** 
 Luo et al. (2023) | Vanilla | 92.0 | 13.6 | 90.3 | 17.5 | 50.9 | 46.2 | 35.3 | 35.2 | 56.4 |
| | FFT with $D^{VQA}$ | 95.6 | 16.3 | 98.3 | 23.1 | 49.8 | 45.2 | 40.9 | 40.2 | 60.4 |
| | FFT with $D^{AUG}$ | 98.5 | 38.2 | 99.0 | 56.1 | 75.0 | 74.8 | 50.0 | 49.7 | 63.9 |
| | + KL | 98.7 | **98.1** | 99.5 | **99.0** | 76.9 | 76.9 | **51.3** | **50.6** | 63.9 |
| | + JS | 98.5 | 97.7 | 98.9 | 98.4 | 78.0 | 76.6 | 50.7 | 50.1 | 64.1 |
| | + RG | **98.9** | 72.5 | 99.1 | 82.2 | 76.0 | 72.6 | 48.3 | 47.6 | 64.2 |
| | + ADV | 98.7 | 72.2 | **99.5** | 85.2 | 76.8 | 76.3 | 49.3 | 48.4 | 64.0 |
| | Ours | 98.4 | 98.0 | 99.2 | 98.3 | **79.0** | **78.3** | 50.2 | 49.7 | **64.8** |
| **Qwen2.5-VL-7B** 
 Luo et al. (2023) | Vanilla | 99.3 | 96.3 | 99.1 | 97.2 | 85.9 | 77.5 | 69.3 | 63.7 | 80.3 |
| | FFT with $D^{VQA}$ | 99.2 | 96.0 | 99.5 | 95.7 | 86.3 | 82.3 | 69.2 | 67.4 | 79.5 |
| | FFT with $D^{AUG}$ | 99.6 | 99.4 | **99.7** | 99.5 | 90.2 | 90.2 | 70.4 | 69.9 | 79.9 |
| | + KL | 99.3 | 99.1 | 99.6 | 99.6 | **92.0** | 91.7 | 71.2 | **70.7** | 80.6 |
| | + JS | 99.5 | 99.4 | 99.7 | 99.3 | 91.6 | **92.1** | **71.5** | 69.9 | 80.3 |
| | + RG | 99.4 | 99.3 | 99.7 | 99.5 | 91.7 | 91.8 | 70.4 | 69.9 | 78.0 |
| | + ADV | 99.4 | 99.2 | 99.6 | 99.5 | 91.7 | 91.8 | 70.4 | 70.0 | 79.9 |
| | Ours | **99.6** | **99.5** | 99.6 | **99.7** | 90.9 | 89.7 | 69.8 | 68.2 | **80.9** |
| *13B Multimodal Models* | | | | | | | | | | |
| **LLaVA-1.5-13B** 
 Liu et al. (2023a) | Vanilla | 95.6 | 73.0 | 97.9 | 77.4 | 65.9 | 63.8 | 51.8 | 50.8 | 66.2 |
| | FFT with $D^{VQA}$ | 94.6 | 72.0 | 97.8 | 80.2 | 67.5 | 64.2 | 52.4 | 52.2 | 65.8 |
| | FFT with $D^{AUG}$ | 98.1 | 96.8 | 96.7 | 96.9 | 81.0 | 78.7 | 52.1 | 51.6 | 67.1 |
| | + KL | 98.3 | 98.0 | 98.8 | 98.5 | 83.0 | **82.6** | 55.7 | 55.1 | 68.0 |
| | + JS | 98.3 | 98.1 | 98.7 | 98.4 | 83.1 | 81.5 | 56.7 | **56.2** | **68.6** |
| | + RG | 98.5 | 98.0 | 98.9 | 98.5 | **83.5** | 82.5 | 55.4 | 55.3 | 66.9 |
| | + ADV | **98.7** | 98.2 | 99.0 | 98.6 | 82.2 | 82.5 | 55.6 | 55.4 | 67.5 |
| | Ours | 98.5 | **98.4** | **99.2** | **98.7** | 83.0 | 82.1 | **56.7** | 56.0 | 68.4 |
| **InstructBlip-13B** 
 Luo et al. (2023) | Vanilla | 95.6 | 73.0 | 97.9 | 77.4 | 65.9 | 63.8 | 51.8 | 50.8 | 65.8 |
| | FFT with $D^{VQA}$ | 95.6 | 8.0 | 97.0 | 11.6 | 58.6 | 55.4 | 43.7 | 42.8 | 59.4 |
| | FFT with $D^{AUG}$ | 98.4 | 9.3 | 99.2 | 13.8 | 82.0 | 80.4 | 52.1 | 51.3 | 65.9 |
| | + KL | 98.5 | **98.3** | 99.1 | 99.2 | 82.5 | 81.4 | **53.4** | **52.5** | 66.2 |
| | + JS | 98.7 | 98.1 | 99.3 | 99.2 | **83.5** | **83.1** | 52.8 | 52.2 | 66.5 |
| | + RG | 98.4 | 97.8 | **99.4** | **99.3** | 80.0 | 76.6 | 50.9 | 50.0 | 66.3 |
| | + ADV | 98.6 | 80.9 | 98.7 | 98.6 | 79.8 | 79.0 | 51.3 | 50.7 | 66.4 |
| | Ours | **98.7** | 97.9 | 98.7 | 98.7 | 83.2 | 81.2 | 52.2 | 51.6 | **66.5** |

Table 7: Detailed multimodal reasoning accuracy (%) on multiple-choice VQA datasets across different ablation study settings with extra models: Qwen2.5-vl-7B, Instructblip-Vicuna-13B. The best accuracy is marked in bold. Overall performance is computed as a weighted average across datasets, with weights proportional to each dataset's test size.

| Model | Method | ScienceQA-IMG | MM-Bench-EN | Seed-Bench-IMG | VQA Overall |
|---|---|---|---|---|---|
| *Qwen2.5-VL Models (Bai et al., 2025)* | | | | | |
| **Qwen2.5-VL-3B** 
 Bai et al. (2025) | Vanilla | 63.0 (100%) | 81.8 (100%) | 72.3 (100%) | 72.5 (100%) |
| | FFT with $D^{VQA}$ | 75.3 (↑19.5%) | 82.0 (↑0.2%) | 74.7 (↑3.3%) | 76.1 (↑5.0%) |
| | FFT with $D^{AUG}$ | 73.6 (↑16.8%) | 81.7 (↓0.1%) | 75.3 (↑4.1%) | 76.2 (↑5.1%) |
| | + KL | 72.8 (↑15.6%) | 80.9 (↓1.1%) | 74.9 (↑3.6%) | 75.9 (↑4.7%) |
| | + JS | 73.4 (↑16.5%) | 82.0 (↑0.2%) | 75.2 (↑4.0%) | 76.2 (↑5.1%) |
| | + RG | 73.4 (↑16.5%) | 81.8 (↑0.0%) | 75.0 (↑3.7%) | 76.0 (↑4.8%) |
| | + ADV | 73.1 (↑16.0%) | 81.7 (↓0.1%) | 75.3 (↑4.1%) | 76.1 (↑5.0%) |
| | Ours | 73.8 (↑17.1%) | 81.5 (↓0.4%) | 75.5 (↑4.4%) | **76.4 (↑5.4%)** |
| **Qwen2.5-VL-7B** 
 Bai et al. (2025) | Vanilla | 85.1 (100%) | 86.6 (100%) | 77.0 (100%) | 80.3 (100%) |
| | FFT with $D^{VQA}$ | 81.4 (↓4.3%) | 86.4 (↓0.2%) | 76.8 (↓0.3%) | 79.5 (↓1.0%) |
| | FFT with $D^{AUG}$ | 83.2 (↓2.2%) | 86.1 (↓0.6%) | 77.0 (-0.0%) | 79.9 (↓0.5%) |
| | + KL | 86.0 (↑1.1%) | 86.7 (↑0.1%) | 77.2 (↑0.3%) | 80.6 (↑0.4%) |
| | + JS | 85.5 (↑0.5%) | 85.9 (↓0.8%) | 77.0 (-0.0%) | 80.3 (-0.0%) |
| | + RG | 81.6 (↓4.1%) | 82.9 (↓4.2%) | 75.5 (↓1.9%) | 78.0 (↓2.8%) |
| | + ADV | 85.2 (↑0.1%) | 85.8 (↓0.9%) | 76.6 (↓0.5%) | 79.9 (↓0.5%) |
| | Ours | 83.9 (↓1.4%) | 86.4 (↓0.2%) | 78.3 (↑1.7%) | **80.9 (↑0.7%)** |
| *Instructblip-Vicuna Models (Luo et al., 2023)* | | | | | |
| **InstructBlip-7B** 
 Luo et al. (2023) | Vanilla | 52.1 (100%) | 65.5 (100%) | 54.8 (100%) | 56.4 (100%) |
| | FFT with $D^{VQA}$ | 61.8 (↑18.6%) | 68.4 (↑4.4%) | 57.6 (↑5.1%) | 60.4 (↑7.1%) |
| | FFT with $D^{AUG}$ | 65.4 (↑25.5%) | 71.1 (↑8.5%) | 61.2 (↑11.7%) | 63.9 (↑13.3%) |
| | + KL | 65.9 (↑26.5%) | 71.7 (↑9.5%) | 60.9 (↑11.1%) | 63.9 (↑13.3%) |
| | + JS | 66.9 (↑28.4%) | 72.3 (↑10.4%) | 60.7 (↑10.8%) | 64.1 (↑13.7%) |
| | + RG | 62.8 (↑20.5%) | 70.9 (↑8.2%) | 62.5 (↑14.0%) | 64.2 (↑13.8%) |
| | + ADV | 66.0 (↑26.7%) | 69.0 (↑5.3%) | 61.8 (↑12.8%) | 64.0 (↑13.5%) |
| | Ours | 64.0 (↑22.8%) | 71.4 (↑9.0%) | 63.0 (↑14.9%) | **64.8 (↑14.9%)** |
| **InstructBlip-13B** 
 Luo et al. (2023) | Vanilla | 65.8 (100%) | 72.1 (100%) | 63.8 (100%) | 65.8 (100%) |
| | FFT with $D^{VQA}$ | 61.7 (↓6.2%) | 68.5 (↓5.0%) | 56.0 (↓12.2%) | 59.4 (↓9.7%) |
| | FFT with $D^{AUG}$ | 66.6 (↑1.2%) | 71.5 (↓0.8%) | 64.0 (↑0.3%) | 65.9 (↑0.2%) |
| | + KL | 67.2 (↑2.2%) | 72.6 (↑0.7%) | 64.0 (↑0.3%) | 66.2 (↑0.6%) |
| | + JS | 67.9 (↑3.2%) | 72.7 (↑0.8%) | 64.2 (↑0.6%) | 66.5 (↑1.1%) |
| | + RG | 65.4 (↓0.6%) | 73.5 (↑1.9%) | 64.3 (↑0.7%) | 66.3 (↑0.8%) |
| | + ADV | 66.1 (↑0.5%) | 74.0 (↑2.6%) | 64.1 (↑0.5%) | 66.4 (↑0.9%) |
| | Ours | 66.2 (↑0.6%) | 73.2 (↑1.5%) | 64.3 (↑0.7%) | **66.5 (↑1.1%)** |
| *LLaVA1.5 Models (Liu et al., 2023a)* | | | | | |
| **LLaVA-1.5-7B** 
 Liu et al. (2023a) | Vanilla | 64.5 (100%) | 64.3 (100%) | 63.4 (100%) | 63.8 (100%) |
| | FFT with $D^{VQA}$ | 61.6 (↓4.5%) | 71.4 (↑11.0%) | 62.4 (↓1.6%) | 64.0 (↑0.3%) |
| | FFT with $D^{AUG}$ | 65.4 (↑1.4%) | 71.1 (↑10.6%) | 63.9 (↑0.8%) | 65.6 (↑2.8%) |
| | + KL | 65.9 (↑2.2%) | 72.5 (↑12.8%) | 64.5 (↑1.7%) | 66.3 (↑3.9%) |
| | + JS | 63.8 (↓1.1%) | 73.5 (↑14.3%) | 63.7 (↑0.5%) | 65.6 (↑2.8%) |
| | + RG | 66.3 (↑2.8%) | 71.8 (↑11.7%) | 63.7 (↑0.5%) | 65.7 (↑3.0%) |
| | + ADV | 66.7 (↑3.4%) | 71.4 (↑11.0%) | 63.6 (↑0.3%) | 65.7 (↑3.0%) |
| | Ours | 67.8 (↑5.1%) | 73.1 (↑13.7%) | 64.6 (↑1.9%) | **66.8 (↑4.7%)** |
| **LLaVA-1.5-13B** 
 Liu et al. (2023a) | Vanilla | 65.8 (100%) | 72.1 (100%) | 64.5 (100%) | 66.2 (100%) |
| | FFT with $D^{VQA}$ | 60.8 (↓7.6%) | 73.6 (↑2.1%) | 64.9 (↑0.6%) | 65.8 (↓0.6%) |
| | FFT with $D^{AUG}$ | 63.5 (↓3.5%) | 75.0 (↑4.0%) | 65.7 (↑1.9%) | 67.1 (↑1.4%) |
| | + KL | 62.5 (↓5.0%) | 74.6 (↑3.5%) | 67.6 (↑4.8%) | 68.0 (↑2.7%) |
| | + JS | 65.8 (↑0.0%) | 74.7 (↑3.6%) | 67.6 (↑4.8%) | **68.6 (↑3.6%)** |
| | + RG | 58.3 (↓11.4%) | 73.5 (↑1.9%) | 67.4 (↑4.5%) | 66.9 (↑1.1%) |
| | + ADV | 57.6 (↓12.5%) | 74.2 (↑2.9%) | 68.3 (↑5.9%) | 67.5 (↑2.0%) |
| | Ours | 62.6 (↓4.9%) | 73.7 (↑2.2%) | 68.4 (↑6.0%) | 68.4 (↑3.3%) |

*Note.* While Qwen2.5-VL was originally instruction-tuned with proprietary in-house data (Bai et al., 2025), our reproduced version uses only publicly available LLaVA instruction-tuning data. Even under this constraint and without access to VQA-specific tuning samples, our models achieve comparable or even better performance across all VQA datasets—highlighting the robustness and effectiveness of our proposed perturbation-consistent fine-tuning strategy.

Table 8: Performance (%) of Vinilla models under modality interference across four datasets. We show accuracy under clean (origin) and various perturbations: Left: Mini-ImageNet and Caltech-101; Right: OpenBookQA and MMLU. UF = Unrelated Facts, MD = Misleading Descriptions, RP = Random Pixels, RI = Real Image, FB & FW = Full Black/White Canvas. (The results are averaged on multiple runs with standard deviation $< 0.2$)

| Model | Mini-ImageNet | | | Caltech-101 | | | Open-Book QA | | | | MMLU | | | |
|---|---|---|---|---|---|---|---|---|---|---|---|---|---|---|
| | Orig | UF | MD | Orig | UF | MD | RP | RI | FB | FW | RP | RI | FB | FW |
| InternVL2-2B (Chen et al., 2024b) | 91.9 | 88.6 | 25.5 | 94.6 | 91.3 | 33.2 | 46.3 | 36.2 | 45.2 | 45.3 | 39.1 | 36.9 | 39.3 | 39.2 |
| LLaVA-1.5-7B (Liu et al., 2023a) | 95.3 | 93.4 | 43.5 | 97.0 | 95.9 | 57.4 | 62.4 | 56.4 | 62.5 | 63.4 | 46.3 | 45.2 | 45.9 | 45.8 |
| LLaVA-Next-7B (Liu et al., 2024) | 93.4 | 90.0 | 28.5 | 97.0 | 93.4 | 31.6 | 54.6 | 52.9 | 55.9 | 55.5 | 45.9 | 45.0 | 45.9 | 45.8 |
| LLaVA-Next-34B (Liu et al., 2024) | 98.0 | 96.3 | 90.2 | 99.1 | 97.3 | 93.6 | 87.7 | 85.7 | 88.4 | 88.0 | 71.5 | 70.9 | 71.5 | 71.5 |
| LLaVA-Next-72B (Liu et al., 2024) | 97.7 | 97.8 | 83.0 | 98.9 | 98.8 | 91.2 | 88.2 | 86.2 | 89.1 | 88.9 | 73.6 | 72.9 | 73.9 | 73.9 |
| LLaVA-Next-110B (Liu et al., 2024) | 98.4 | 98.3 | 93.3 | 98.4 | 98.3 | 93.1 | 89.5 | 89.2 | 89.9 | 89.7 | 73.5 | 73.0 | 73.9 | 73.9 |
| LLaVA-1.5-13B (Liu et al., 2023a) | 95.6 | 94.1 | 73.0 | 97.9 | 97.1 | 77.4 | 65.9 | 63.8 | 68.0 | 69.1 | 51.8 | 50.8 | 52.7 | 52.7 |
| InstructBlip-7B (Luo et al., 2023) | 92.0 | 87.1 | 13.6 | 90.3 | 90.2 | 17.5 | 50.8 | 46.2 | 50.9 | 50.7 | 36.3 | 35.2 | 36.2 | 36.7 |
| InstructBlip-13B (Luo et al., 2023) | 93.0 | 81.6 | 50.9 | 94.1 | 82.7 | 51.0 | 44.7 | 39.4 | 45.6 | 46.1 | 40.1 | 37.4 | 40.5 | 42.9 |
| QwenVL2-2B (Wang et al., 2024b) | 98.7 | 98.6 | 75.8 | 99.1 | 99.1 | 66.8 | 59.6 | 54.7 | 63.4 | 63.8 | 49.5 | 44.6 | 49.7 | 50.0 |
| QwenVL2.5-3B (Bai et al., 2025) | 98.9 | 98.5 | 94.9 | 98.8 | 99.0 | 94.4 | 79.9 | 74.6 | 80.0 | 79.7 | 63.5 | 61.1 | 64.0 | 63.6 |
| QwenVL2-7B (Wang et al., 2024b) | 99.0 | 99.1 | 96.3 | 99.6 | 99.5 | 97.6 | 82.5 | 80.9 | 83.7 | 83.2 | 66.9 | 65.3 | 67.7 | 67.8 |
| QwenVL2.5-7B (Bai et al., 2025) | 99.3 | 99.3 | 96.5 | 99.3 | 99.4 | 96.8 | 86.2 | 80.9 | 86.3 | 86.4 | 69.5 | 67.6 | 69.1 | 69.1 |

## C.1 Perturbation-based Evaluation Experiment Results

We conduct a controlled perturbation-based evaluation across various MLLMs, as shown in Tab. 8. Our results reveal that both vision and language tasks are vulnerable to cross-modal interference. In vision classification tasks, misleading textual descriptions (e.g., text contradicting image content) lead to severe performance drops. For example, InternVL2-2B and InstructBLIP-7B on Mini-ImageNet drop from 91.9% to 25.5% and from 92.0% to 13.6%, respectively. Conversely, for language tasks such as OpenBookQA and MMLU, irrelevant visual inputs—particularly semantically unrelated real images—also degrade performance. LLaVA-1.5-7B drops from 46.3% to 45.2% on MMLU, while InstructBLIP-13B sees over 5 points of degradation.

A consistent trend is that larger models exhibit greater robustness. Models like LLaVA-1.5-13B and QwenVL2.5-7B maintain high accuracy across all perturbation types—e.g., QwenVL2.5-7B sustains over 96% on Mini-ImageNet with misleading text and over 86% on OpenBookQA with irrelevant images—indicating improved modality disentanglement and reduced sensitivity to spurious correlations. Nonetheless, performance still degrades relative to clean inputs, highlighting that interference effects remain non-negligible even in stronger models.

We further observe a clear scaling trend within the LLaVA-Next family. As model size increases from 7B to 34B, 72B, and 110B, performance under perturbations steadily improves, reflecting stronger representation power and enhanced robustness to spurious cues. For instance, LLaVA-Next-7B achieves 90.3% on Mini-ImageNet (Orig) but drops to 28.5% with misleading descriptions, whereas LLaVA-Next-110B maintains 98.4% and 93.3% under the same conditions. Similarly, on MMLU, accuracy under irrelevant real images increases from 45.9% (7B) to 73.6% (72B). **These results confirm that scaling up helps mitigate modality interference. However, such gains come at substantial computational and resource costs, and the improvements remain incremental relative to the clean–perturbed gap.** This underscores that scaling alone is insufficient, and more targeted interventions—such as our proposed framework—are necessary for robust cross-modal reasoning.

## C.2 Experiment Results on InstructBlip-Vicuna-13b and QwenVL-2.5-7b

To enable a more equitable comparison with existing multimodal models, we extend our method to two additional backbones: InstructBLIP-Vicuna-13B and QwenVL-2.5-7B. As shown in Tab. 7, our approach consistently improves performance across multiple VQA benchmarks, even under different instruction-tuning conditions, demonstrating its robustness and general applicability.

## C.3 DISCUSSION ON KL&JS USE FOR CONSISTENCY REGULARIZATION

Although both KL and JS divergence serve as effective objectives for consistency regularization, we find that JS consistently achieves slightly better results across most settings. Specifically, in both unimodal tasks (e.g., Mini-ImageNet, MMLU) and multimodal reasoning benchmarks (e.g., ScienceQA, SeedBench), JS-regularized models consistently outperform their KL counterparts by a small but observable margin. This trend holds across different model backbones and training configurations, including our final unified method (see "Ours" rows in Table 5 and 7). This suggests a marginal advantage of JS regularization in enhancing model robustness.

## C.4 EVALUATING CHAIN-OF-THOUGHT PROMPTING FOR MODALITY INTERFERENCE MITIGATION

To further investigate the potential of prompt-based methods in mitigating modality interference, we conduct additional experiments using Chain-of-Thought Wei et al. (2022) style prompting. This approach aims to encourage structured reasoning by guiding the model through an explicit reasoning process before producing its final answer.

Specifically, we prepend the following CoT prompt to each input question:

```
Let's think step by step:
1.  What information does the image provide?
2.  What is the question asking?
3.  Are there any misleading parts?
4.  Now give your final answer.  Only write the final answer
on a separate line like:  ``Answer: B''
```

Results are presented in Table 9. The results suggests that structured reasoning alone cannot resolve the interference problem, as the issue stems from misaligned cross-modal representations rather than shallow reasoning steps.

Table 9: Accuracy (%) under different interference settings across tasks and models. Each task includes original inputs and multiple types of perturbations.

| Model | Method | Mini-ImageNet | | | Caltech-101 | | | OpenBookQA | | | | MMLU | | | |
|---|---|---|---|---|---|---|---|---|---|---|---|---|---|---|---|
| | | Orig | UF | MD | Orig | UF | MD | RP | RI | FB | FW | RP | RI | FB | FW |
| vanilla | llava-1.5-7b | 95.3 | 93.4 | 43.5 | 97.0 | 95.9 | 57.4 | 62.4 | 56.4 | 62.5 | 63.4 | 46.3 | 45.2 | 45.9 | 45.8 |
| CoT | llava-1.5-7b | 81.7 | 70.9 | 28.9 | 80.9 | 71.8 | 36.0 | 38.8 | 38.9 | 41.0 | 41.0 | 39.6 | 38.7 | 40.2 | 40.4 |
| Ours | llava-1.5-7b | 98.6 | 98.4 | 98.7 | 99.3 | 98.9 | 99.3 | 81.8 | 81.0 | 81.7 | 81.7 | 51.5 | 50.9 | 51.5 | 51.4 |
| vanilla | llava-1.5-13b | 95.6 | 94.1 | 73.0 | 97.9 | 97.1 | 77.4 | 65.9 | 63.8 | 68.0 | 69.1 | 51.8 | 50.8 | 52.7 | 52.7 |
| CoT | llava-1.5-13b | 92.9 | 85.9 | 62.8 | 96.6 | 92.1 | 67.8 | 55.6 | 53.0 | 56.5 | 57.3 | 47.3 | 45.5 | 47.3 | 47.8 |
| Ours | llava-1.5-13b | 98.7 | 97.9 | 98.0 | 98.7 | 98.7 | 98.8 | 83.2 | 81.2 | 83.8 | 83.0 | 52.2 | 51.6 | 52.3 | 53.4 |
| vanilla | qwen2.5-vl-3b | 98.9 | 98.5 | 94.9 | 98.8 | 99.0 | 94.4 | 79.9 | 74.6 | 80.0 | 79.7 | 63.5 | 61.1 | 64.0 | 63.6 |
| CoT | qwen2.5-vl-3b | 92.3 | 96.8 | 88.2 | 94.7 | 96.1 | 86.0 | 61.0 | 52.8 | 61.3 | 61.1 | 51.5 | 48.7 | 51.6 | 51.4 |
| Ours | qwen2.5-vl-3b | 99.3 | 99.2 | 99.2 | 99.7 | 99.7 | 99.5 | 86.7 | 86.4 | 86.6 | 86.6 | 64.8 | 64.5 | 65.0 | 65.1 |
| vanilla | qwen2.5-vl-7b | 99.3 | 99.3 | 96.3 | 99.1 | 98.9 | 97.2 | 85.9 | 77.5 | 85.8 | 86.0 | 69.3 | 63.7 | 68.9 | 68.9 |
| CoT | qwen2.5-vl-7b | 99.1 | 98.9 | 91.4 | 98.5 | 98.5 | 92.5 | 77.8 | 68.8 | 77.3 | 77.6 | 61.5 | 57.4 | 60.8 | 60.9 |
| Ours | qwen2.5-vl-7b | 99.6 | 99.5 | 99.5 | 99.6 | 99.6 | 99.7 | 90.9 | 89.7 | 90.8 | 91.0 | 69.8 | 68.2 | 69.8 | 70.0 |

## C.5 EVALUATION ON FREE-FORM VQA

To further assess the generalizability of our method beyond multiple-choice VQA tasks, we evaluate on TextVQA (Singh et al., 2019), a free-form generative visual question answering dataset that requires reasoning over both textual and visual content in natural images. We follow the benchmark adopted by LLaVA Liu et al. (2023b), which evaluates a model's ability to both recognize textual characters within images and effectively handle noisy outputs generated by OCR systems.

Following standard evaluation protocols and existing MLLM baselines (e.g., LLaVA, Qwen2.5-VL, InstructBLIP), we report model performance averaged over multiple runs (standard deviation < 0.4). Results are presented in Table 10.

Our method achieves consistent improvements across most model families, indicating its effectiveness not only in MCQA scenarios but also in open-ended multimodal reasoning settings.

Table 10: Accuracy (%) on the TextVQA dataset across different model families.

| Setting | Model | TextVQA |
|---------|-------|---------|
| Vanilla | InstructBLIP-Vicuna-7B | 19.7 |
| Ours | InstructBLIP-Vicuna-7B | **32.4** |
| Vanilla | LLaVA-1.5-7B | 49.8 |
| Ours | LLaVA-1.5-7B | **51.2** |
| Vanilla | Qwen2.5-VL-7B | 81.4 |
| Ours | Qwen2.5-VL-7B | **84.8** |

## C.6 IMPROVING GENERALIZABILITY THROUGH ADVERSARIAL TRAINING

To assess the generalization benefits of adversarial training, we evaluate model robustness under two types of out-of-distribution (OOD) perturbations at test time:

- **Document OCR noise:** Real-world noisy OCR snippets are sampled from the FUNSD dataset (Jaume et al., 2019) and inserted as irrelevant textual distractors into visual classification tasks (Mini-ImageNet, Caltech-101).
- **Unrelated screenshots:** Unrelated UI screenshots are drawn from the RICO dataset (Deka et al., 2017) and added as visual distractors to language-dominant VQA tasks (OpenBookQA, MMLU).

Each experiment is repeated multiple times, and we report average accuracy across runs (standard deviation < 0.1). Results are presented in Table 11. Across all model scales and task types, adversarial training consistently improves robustness to both types of perturbations. These findings indicate that the additional training overhead introduced by adversarial perturbation is well-justified by the improved generalization to unseen distribution shifts—a desirable property for reliable deployment in real-world settings.

Table 11: Accuracy (%) on original and perturbed inputs. OCR snippets are inserted into image classification tasks, and RICO screenshots are added to VQA tasks.

| Setting | Model | Mini-ImageNet | | Caltech-101 | | OpenBookQA | | MMLU | |
|---------|-------|--------|-----|--------|-----|-----------|------------|-----------|------------|
| | | Origin | OCR | Origin | OCR | RandPixels | Screenshot | RandPixels | Screenshot |
| vanilla | instructblip-vicuna-7b | 92.0 | 81.4 | 90.3 | 83.6 | 50.9 | 40.0 | 35.3 | 34.9 |
| SFT | instructblip-vicuna-7b | 98.5 | 95.7 | 99.0 | 98.4 | 75.0 | 74.4 | 50.0 | 49.2 |
| SFT + ADV | instructblip-vicuna-7b | 98.7 | 97.5 | 99.5 | 98.8 | 76.8 | 76.2 | 49.3 | 49.3 |
| Ours | instructblip-vicuna-7b | 98.4 | 97.3 | 99.2 | 99.0 | 79.0 | 77.8 | 50.2 | 49.2 |
| vanilla | llava-1.5-7b | 95.3 | 88.9 | 97.0 | 92.8 | 62.4 | 50.2 | 46.3 | 44.8 |
| SFT | llava-1.5-7b | 98.2 | 98.0 | 98.5 | 98.0 | 78.6 | 78.0 | 51.1 | 50.6 |
| SFT + ADV | llava-1.5-7b | 98.7 | 98.2 | 98.7 | 98.4 | 81.7 | 81.3 | 50.6 | 51.3 |
| Ours | llava-1.5-7b | 98.6 | 98.2 | 99.3 | 99.0 | 81.8 | 81.2 | 51.5 | 51.3 |
| vanilla | qwen2.5-vl-7b | 99.3 | 99.2 | 99.1 | 99.2 | 85.9 | 69.3 | 69.3 | 57.0 |
| SFT | qwen2.5-vl-7b | 99.6 | 99.5 | 99.7 | 99.3 | 90.2 | 85.8 | 70.4 | 63.7 |
| SFT + ADV | qwen2.5-vl-7b | 99.4 | 99.5 | 99.6 | 99.6 | 91.7 | 91.5 | 70.4 | 69.7 |
| Ours | qwen2.5-vl-7b | 99.6 | 99.5 | 99.6 | 99.5 | 90.9 | 90.7 | 69.8 | 69.7 |

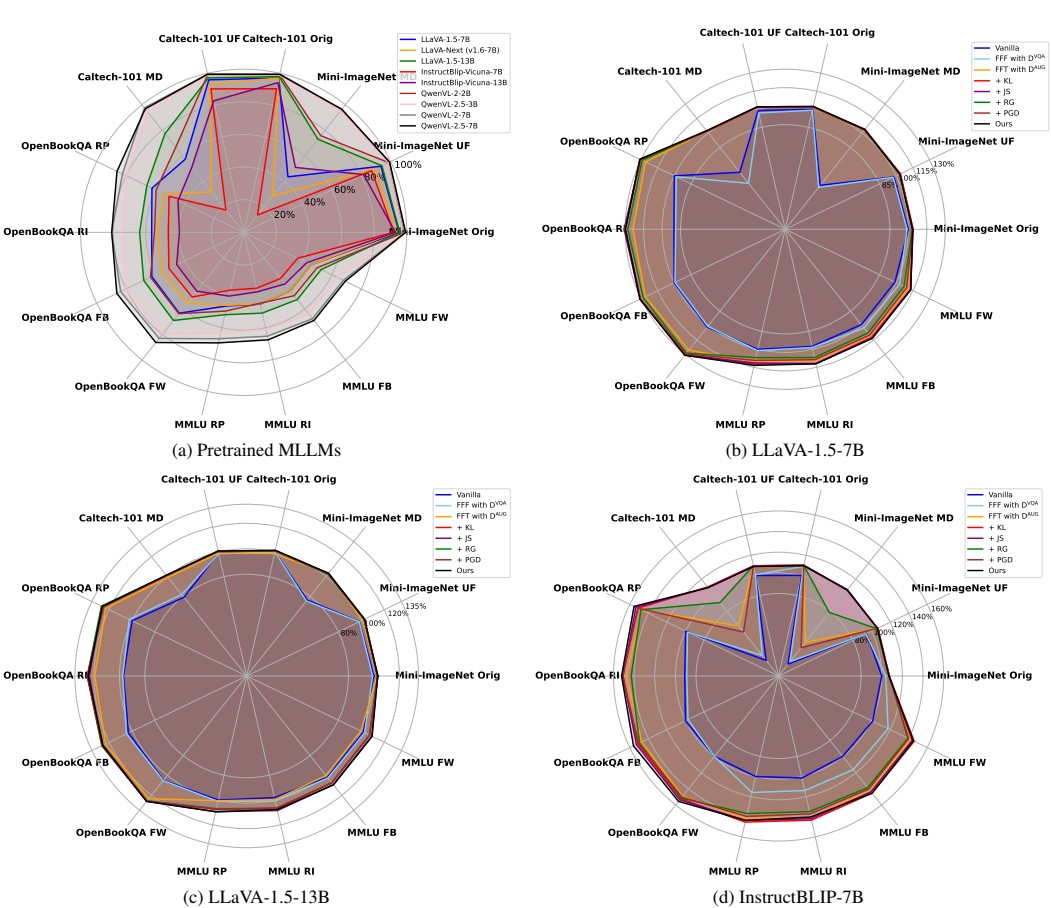

Figure 4: Task-wise robustness under perturbation. Each radar chart shows model accuracy (%) across Mini-ImageNet, Caltech-101 (image-heavy) and OpenBookQA, MMLU (text-heavy) under various perturbations. (a) uses raw accuracy of different pretrained MLLMs directly. (b–d) are normalized relative accuracy of each MLLMs. (We normalize each absolute accuracy into relative accuracy, which refers to absolute tested accuracy / accuracy of vanilla MLLMs in origin setting without perturbation.)

Table 12: Hyperparameter Settings Example

| Category | Setting |
|----------|---------|
| *Model and Training Strategy* | |
| Base Model | LLaVA-1.5-7B |
| Finetune Type | Full |
| Adversarial Type | PGD-alike |
| Step size $\alpha$ | 0.1 |
| Epsilon-hall $\epsilon$ | 0.001 |
| Avdersarial Training Steps $T$ | 2 |
| Consistency Regularization Type | JS |
| Loss Weight $\lambda_{\text{consistency}}$ | 0.01 |
| Temperature $\tau$ | 1 |
| *Optimization* | |
| Epochs | 1 |
| Batch Size per GPU $|\mathcal{B}|$ | 8 |
| Img/Text Ratio | 0.25/0.25 |
| Learning rate | $2 \times 10^{-5}$ |

Table 13: Dataset Statistics

| Dataset | Train | Test |
|---------|-------|------|
| Mini-ImageNet | 4935 | 2000 |
| Caltech-101 | 8124 | 1020 |
| OpenBookQA | 4957 | 1000 |
| MMLU | 7M | 5469 |
| LLaVA-Instruct | 624610 | - |
| TextCaps | 109765 | - |
| MMBench-EN | - | 4377 |
| ScienceQA-IMG | - | 4114 |
| SeedBench-IMG | - | 14243 |

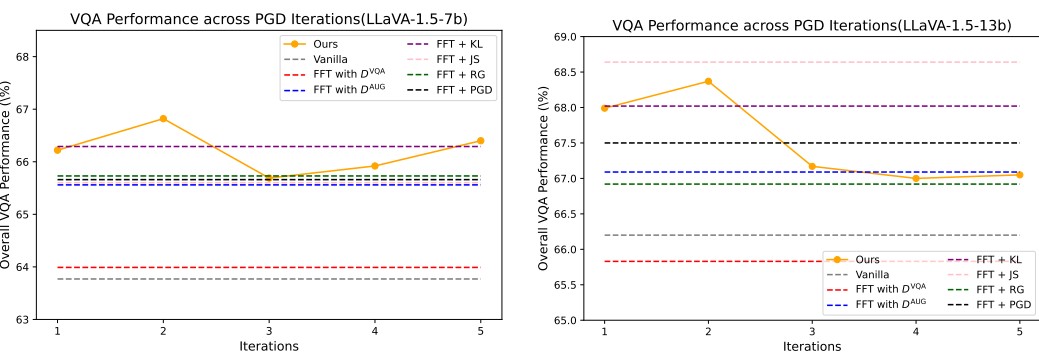

Figure 5: Comparison of VQA performance across adversarial training iterations for different model sizes.

## D  HYPER-PARAMETER SETTING AND TRAINING DETAILS

Please see Tab. 12 for more details.

### D.1  PARAMETER ANALYSIS WITH ADVERSARIAL TRAINING ITERATIONS

To investigate the effect of adversarial strength on model performance, we vary the number of adversarial training iterations from 1 to 5 and evaluate the resulting VQA accuracy. As shown in Fig. 5, both LLaVA-1.5-7B and LLaVA-1.5-13B models benefit from adversarial consistency training, with performance peaking at 2-step adversarial training (66.82% and 68.37%, respectively). Notably, excessive iterations (e.g., 4 or 5 steps) may lead to slight degradation, especially in larger models, likely due to over-perturbation and optimization difficulty.

These findings suggest that a moderate adversarial training setting (2 steps with $\epsilon$=1e-3 and $\alpha$=0.1 in LLaVA-1.5-7b, $\epsilon$=1e-4 and $\alpha$=0.1 in LLaVA-1.5-13b) offers an optimal balance between robustness and training stability, and that model size influences sensitivity to adversarial signal strength.

## E EXPERIMENTS COMPUTE RESOURCES

All experiments were conducted on $8\times$A100 GPUs using DeepSpeed ZeRO-3 (Contributors, 2024) with CPU offloading.

To quantify the computational overhead introduced by our adversarial training, we provide both theoretical FLOPs analysis and empirical wall-clock training time across model scales.

In the standard supervised fine-tuning (SFT) setting, the FLOPs per batch can be approximated as:

$$\text{FLOPs}_{\text{SFT}} \approx B_s \cdot (f_{\text{LLM}} + b_{\text{LLM}}), \tag{10}$$

where $B_s$ is the batch size, $f_{\text{LLM}}$ denotes the FLOPs of a forward pass through the LLM, and $b_{\text{LLM}}$ the backward pass. In our adversarial training, each sample undergoes $N$ adversarial training steps, each requiring an additional forward pass through the *frozen* LLM. Since gradients are computed only with respect to input embeddings via `torch.autograd.grad`, the overhead is minimal and thus ignored. After perturbation, clean and adversarial inputs are concatenated, resulting in a forward cost of $2B_s \cdot f_{\text{LLM}}$, followed by one backward pass. The total cost becomes:

$$\text{FLOPs}_{\text{ADV+SFT}} = B_s \cdot (Nf_{\text{LLM}} + 2f_{\text{LLM}} + b_{\text{LLM}}). \tag{11}$$

The relative overhead compared to vanilla SFT is:

$$\frac{Nf + 2f + b}{f + b}. \tag{12}$$

Assuming $b_{\text{LLM}} \approx 2f_{\text{LLM}}$, this simplifies to:

$$\frac{N + 4}{3}. \tag{13}$$

For our default $N = 1$, the theoretical FLOPs increase to approximately $1.66\times$ that of SFT.

We further report the actual training time across model scales, as shown in Table 14.

Table 14: Training time (in hours) across model scales. $\Delta$ denotes additional overhead per adversarial training iteration step.

| Model | SFT | SFT + KL | SFT + ADV | Ours (ADV + KL) |
|---|---|---|---|---|
| Qwen2.5-VL-3B | 1.5h | 1.5–1.75h | 3–3.5h ($\Delta = 0.5$h) | 3.5–4h ($\Delta = 0.5$h) |
| LLaVA-1.5-7B | 4h | 4–4.5h | 6–6.5h ($\Delta = 0.5$h) | 6–6.5h ($\Delta = 0.5$h) |
| LLaVA-1.5-13B | 10h | 10–11h | 13–14h ($\Delta = 1$h) | 13–14h ($\Delta = 1$h) |

Although the theoretical FLOPs suggest a $\sim$66% increase in cost when $N = 1$, the actual wall-clock time increase is much smaller. This is because our designed adversarial training leverages forward-only passes over frozen LLMs, avoiding costly backward and optimizer updates. As a result, the added runtime remains modest even on large models (e.g., only +2.5h for LLaVA-13B). Moreover, KL consistency training introduces negligible overhead compared to SFT.

## F LIMITATIONS

Our analysis of modality interference is conducted from a coarse-grained perspective, primarily categorizing tasks into image-heavy and text-heavy types. A more fine-grained investigation—such as dynamic attention tracking—could provide deeper insights into how MLLMs rely on or ignore specific modalities during reasoning.

Moreover, while our perturbation strategies (e.g., unrelated facts, misleading descriptions, irrelevant images) effectively reveal failure modes of current MLLMs, they remain heuristic and task-specific. Designing perturbations is, by nature, an open-ended process—one can always propose new forms of misleading inputs. Thus, an ultimate goal is to develop perturbation-agnostic methods that improve robustness without requiring exhaustive enumeration of possible attacks.

While our use of adversarial training represents a strong and generalizable perturbation strategy, it still operates within a defined input space (e.g., embedding-level noise bounded by $L_\infty$ norms). Hence, adversarial perturbation should be viewed as a practical but partial solution rather than a comprehensive defense. Developing mechanisms that generalize across both semantic and modality perturbations remains an open and challenging direction.

# G  BROADER IMPACTS

This work investigates the limitations of current multimodal large language models in reasoning across modalities and proposes methods to mitigate modality interference—a concrete failure case of cross-modality competency. By improving the robustness and alignment behavior of MLLMs, our approach may benefit a variety of downstream applications that rely on accurate visual-linguistic understanding, including education, accessibility tools (e.g., visual question answering for blind users), and scientific multimodal reasoning tasks.

Our findings also highlight the hidden risks of over-relying on irrelevant modality signals, which can degrade performance or lead to misleading predictions. Making such failure modes measurable and diagnosable can support safer deployment and more transparent evaluation of MLLMs in practice.

On the other hand, techniques such as adversarial perturbation may be dual-use. While our implementation uses perturbations to improve model alignment, similar strategies could be misused to manipulate model behavior. To reduce such risks, we restrict all experiments to open-source academic models and do not include harmful or sensitive content in training or evaluation. We encourage future work to further assess modality interference in safety-critical contexts and to investigate alignment-aware perturbation techniques with explicit safety constraints.

# H  THE USE OF LARGE LANGUAGE MODELS

We used large language models only to edit the manuscript for clarity, grammar, and academic style. No part of the research design, data analysis, or scientific content relied on language models, and the authors retain full responsibility for the paper's ideas and conclusions.

