# OpenReview forum: "Diagnosing and Mitigating Modality Interference in Multimodal Large Language Models"
_ICLR.cc/2026/Conference — Submitted to ICLR 2026_

### Official Review · Reviewer_zGvT · 2025-10-31

**Soundness:** 3
**Presentation:** 3
**Contribution:** 3
**Rating:** 6
**Confidence:** 3

**Summary:**

This paper introduces a framework to diagnose and mitigate modality interference in multimodal large language models (MLLMs)—a phenomenon where irrelevant or misleading modality signals degrade model performance. The authors define the broader cross-modality competency problem, identifying modality interference as a concrete instance. They propose (1) a perturbation-based causal evaluation that quantifies interference by injecting irrelevant signals into one modality, and (2) a fine-tuning strategy combining heuristic and adversarial perturbations with a consistency regularization objective. Experiments on image-heavy, text-heavy, and multimodal tasks (Mini-ImageNet, Caltech-101, OpenBookQA, MMLU, ScienceQA, MM-Bench, Seed-Bench) demonstrate significant robustness gains and improved unimodal reasoning without harming multimodal performance. The paper is technically solid and the framing is clear, though the conceptual advance is moderate.

**Strengths:**

1. The identification of modality interference as a measurable phenomenon and its connection to cross-modality competency is insightful.
2. The perturbation-based causal evaluation is well designed and empirically grounded, revealing meaningful vulnerability patterns.
3. The fine-tuning framework combining heuristic and adversarial perturbations with consistency regularization is practically effective.
4. The experiments are comprehensive across datasets, model sizes, and modalities, showing consistent improvements in robustness and generalization.

**Weaknesses:**

1. The theoretical depth is limited. The work largely integrates known ideas from causal probing and adversarial robustness without deeper theoretical analysis.
2. The causal effect metric ($\delta_{cp}$) is intuitive and a bit heuristic. It only captures prediction flips, not probabilistic changes or feature-level shifts.
3. The perturbation design may unintentionally change semantic content rather than purely isolate modality relevance.
4. The method introduces additional computation from adversarial training, and the efficiency trade-off is not discussed.
5. The paper emphasizes robustness metrics but provides little mechanistic insight into why the proposed perturbation strategy improves cross-modality alignment.

**Questions:**

1. How sensitive are the robustness gains to the perturbation strength \epsilon and the number of adversarial steps?
2. Can the proposed approach generalize to generative tasks or multimodal reasoning beyond multiple-choice formats?
3. How does the model behave under simultaneous perturbations in both modalities?
4. Could the causal effect be defined more continuously (e.g., using KL divergence on prediction distributions) to better quantify partial modality reliance?
5. Have the authors compared to other causal fine-tuning strategies such as counterfactual supervision or gradient orthogonalization?

---

> ### Author Response · Authors · 2025-11-24
> **Rebuttal-part 1**
>
> We sincerely thank the reviewer for the constructive feedback.
>
> ## Theoretical Depth
> We respectfully clarify that our contribution goes beyond combining existing ideas; we provide the **first formalization of Modality Interference** using Structural Causal Models, which serves as the theoretical backbone connecting our entire framework. This causal perspective ensures that every design choice is theoretically grounded and well-motivated: it first elevates the problem from generic robustness to a rigorous definition of **Cross-Modality Competency**, unifying disparate failure modes (e.g., catastrophic forgetting, knowledge conflict) as the non-zero causal effect of irrelevant variables. Furthermore, as noted in the paper, this theory explicitly guides our methodology from diagnosis to mitigation—our diagnostic metric $\delta_{cp}$ is derived to empirically estimate the **Total Causal Effect** (TCE) of spurious pathways, while our mitigation strategy is designed to control the **Direct Causal Effect** (DCE) of latent representations. By maximizing non-causal shortcuts via adversarial perturbations and enforcing consistency, we mechanistically sever these spurious causal paths, compelling the model to ground its reasoning solely on the task-relevant modality.
>
> ## Causal Effect Metric - More Continuous Definition
> We respectfully clarify that our framework strategically utilizes **both discrete and continuous measures** to ensure distinct yet complementary goals. For diagnosis, we adopt the prediction flip rate ($\delta_{cp}$) to rigorously align with the standard definition of Average Treatment Effect (ATE) in causal inference [1], capturing the critical "functional failure point" where interference becomes strong enough to alter the final decision. Conversely, to address the reviewer's valid concern regarding partial modality reliance, our mitigation strategy explicitly employs continuous Consistency Regularization (e.g., KL/JS divergence in Eq. 8) during optimization. This design ensures that the model is trained to resist subtle, feature-level probabilistic shifts, thereby optimizing for continuous stability while evaluating against strict functional reliability.
>
> ## Perturbation change Semantics & Simultaneous Perturbations
> Semantic Integrity is Mathematically Guaranteed via Masking: We respectfully clarify that our perturbation design cannot unintentionally alter the task-relevant semantic content due to our novel **Modality-Specific Perturbation Masking mechanism**. As defined in Eq. 6 ($E = E + \delta \odot M$), we explicitly apply a binary mask $M$ that restricts adversarial perturbations strictly to the irrelevant modality tokens. For example, for an image-heavy task, the mask ensures that the image embeddings $Z_I$ remain strictly unperturbed (identity mapping). We only optimize perturbations on the text embeddings $Z_T$. Therefore, the semantic content required to solve the task is mathematically preserved. The goal is precisely to train the model to ignore these semantic shifts in the irrelevant channel while maintaining consistency with the protected, relevant modality.
>
> Additionally, We implemented the "No Mask" variant, where adversarial perturbations are applied to all modalities across all datasets (including VQA).
>
> |Model|Caltech-101(origin)|Caltech-101(MisleadText)|Caltech-101(OCRText)|OpenbookQA(RandomPixel)|OpenbookQA(RealImage)|OpenbookQA(Screenshots)|VQAOverall|
> |-|-|-|-|-|-|-|-|
> |llava7b-vanilla|97|57.4|92.8|62.4|56.4|50.2|63.8|
> |llava7b-Ours(No Mask)|98.1|96.3|93.5|79.5|78.1|62.7|61.5|
> |llava7b-Ours|99.3|98.9|99|81.8|81|81.8|66.8|
>
> As shown in table, (1)no mask training still partially mitigates modality interference—because the combination of consistency regularization and simple heuristic perturbations already teaches the model useful invariances, and adversarial training itself improves generic robustness(e.g. Screenshots 50.2 → 62.7). (2)The mask is crucial for **OOD generalization**: No mask leads to collapses on unseen perturbations (e.g., Screenshots: 81.8 → 62.7). (3) The mask prevents **semantic destruction** during VQA training: when all the token embeddings are perturbed, the model cannot preserve multimodal semantics, leading to a notable drop in VQA accuracy (66.8 vs. No Mask 61.5). Although, simultaneous perturbations fall outside the scope of our causal formulation and research questions. This risks destroying the necessary semantic content required for the ground truth, transforming the problem from "Competency" to "Denoising".
>
> [1] Causal diagrams for empirical research. Judea Pearl

---

> ### Author Response · Authors · 2025-11-24
> **Rebuttal-part 2**
>
> ## Cost of Adversarial Training
> We respectfully point out that a comprehensive analysis of the computational efficiency and trade-offs is provided in Appendix E ("Experiments Compute Resources"). We apologize if this was not prominent enough in the main text. The results demonstrate that a modest  increase in training overhead (detailed in Appendix E) yields substantial gains (Table 4, 11, Appendix C6, unseen dataset evaluation), such as improved performance from 50.2% to 81.2% on screenshot perturbation on llava1.5-7b.
>
> ## Mechanistic insights
> We clarify that the improvement in cross-modality alignment stems from a mechanistic process of "Causal Path Purification." Mechanistically, modality interference arises when models rely on spurious correlations—such as language priors or salient but irrelevant visual objects—acting as non-causal shortcuts. Our adversarial perturbations specifically target and maximize these shortcuts, while consistency regularization acts as a constraint that explicitly severs these spurious causal paths. By suppressing these "easy" but erroneous shortcuts, the optimization process is compelled to restore the model's Cross-Modality Competency: it forces the model to fairly evaluate and integrate signals from all modalities based on causal relevance rather than bias. This results in a higher "signal-to-noise" ratio in latent representations, directly translating to improved visual grounding and general multimodal reasoning. We thank the reviewer for prompting this deeper mechanistic analysis and will incorporate this discussion into the revision to strengthen the paper's theoretical grounding.
>
> ## Sensitivity Analysis
> We have empirically analyzed this sensitivity in Appendix D.1 and Figure 5. As shown in the results, our method yields consistent interference-mitigation improvements across a range of iterations (1–5 steps) for both LLaVA-7B and 13B models. While performance peaks at 2 steps, the gains remain robust even at higher iterations, aligning with established findings in adversarial training literature [2, 3] where "early stopping" or few-step attacks often suffice for perturbation.
>
> Regarding $\epsilon$, we treat it as a constrained design parameter dictated by the embedding space geometry. The $\epsilon$-ball represents the maximum magnitude of noise allowed, as it must be large enough to generate effective gradient signals but bounded to strictly preserve the semantic identity of tokens (the "semantic-safe" range). The fact that our models converge stably and show consistent improvements across different scales with these standard $\ell_\infty$ settings suggests the method is robust within the reasonable "semantic-safe" range of $\epsilon$ (eg. 10^-3 for llava7b).
>
> ## Generalize to generative tasks
> We have indeed evaluated our method on generative tasks beyond multiple-choice formats. As detailed in Appendix C.5 and Table 10, we tested our approach on TextVQA, a representative benchmark that requires the model to generate open-ended text responses based on visual grounding. The results demonstrate that our method consistently improves performance across different models (e.g. 19.7% to 32.4% (+12.7%) on insBlip7b), indicating that the proposed "Causal Path Purification" mechanism effectively enhances the model's ability to ground generation in relevant multimodal signals, regardless of the output format.
>
> ## Other Causal FT Strategies
> We thank the reviewer for suggestion. We respectfully clarify that while our method is not a direct implementation of existing counterfactual supervision pipelines, it follows the same underlying causal principle. By constructing perturbed inputs while keeping the task label unchanged, we explicitly generate counterfactual samples that alter irrelevant causal variables. Our consistency regularization then enforces prediction invariance to these counterfactual interventions, corresponding to penalizing the DCE of irrelevant modalities. In this sense, our framework operationalizes counterfactual supervision in the embedding space of MLLMs, where modality-specific masking and raw-gradient perturbations allow finer control than prior feature-level formulations[4].
>
> Methods such as PCGrad [5]  address spurious correlations by enforcing gradient orthogonality to bias-related directions. While conceptually appealing, applying such strategies to 7B–13B multimodal LLMs is impractical due to scalability: Orthogonalization requires storing multiple gradient vectors or performing per-layer projections, which is prohibitively expensive for billion-parameter models both in memory and computation, while our adversarial perturbation operates only at the input-token embedding level.
>
> [2] Towards Deep Learning Models Resistant to Adversarial Attacks.
>
> [3] Virtual Adversarial Training: A Regularization Method for Supervised and Semi-Supervised Learning
>
> [4] Counterfactual VQA: A Cause-Effect Look at Language Bias
>
> [5] Gradient Surgery for Multi-Task Learning

---

> ### Author Response · Authors · 2025-11-26
>
> Dear Reviewer zGvT,
>
> Thank you again for your detailed and thoughtful review. We’ve provided our rebuttal and would really appreciate any further feedback you may have. Looking forward to your thoughts and discussion!
>
> Sincerely,
>
> The Authors

---

### Official Review · Reviewer_GRpQ · 2025-11-01

**Soundness:** 3
**Presentation:** 3
**Contribution:** 2
**Rating:** 4
**Confidence:** 4

**Summary:**

The paper proposes the cross-modality competency problem where a multimodal large language model (MLLM) fails to evaluate all modalities. A concrete example of this problem is modality interference, where MLLMs often use spurious information from irrelevant modalities when they are expected to rely solely on one-modality data. As a result, MLLMs often underperform on pure visual recognition and textual reasoning. The paper designs a perturbation-based experiment to verify and quantify this problem. Then, it proposes to fine-tune MLLMs with perturbation-based data augmentations. Experiments on image-heavy, text-heavy and multimodal tasks  and multiple model families verify the effectiveness of the proposed approach in boosting unimodal reasoning ability while enhancing performance on multimodal tasks.

**Strengths:**

- The paper demonstrates that MLLMs are not robust under modality interference where different modalities are not aligned and only one modality is relevant to the task. This highlights an important robustness issue in MLLMs.

- The paper shows that using modality misaligned data for fine-tuning can mitigate modality interference and is effective in boosting both unimodal and multimodal reasoning abilities.

- Experiments show that the proposed method is effective with different model families in different scales.

- The paper is well-written and easy to follow.

**Weaknesses:**

- The technical contributions of the paper are limited. The proposed perturbation-based data augmentations are not novel, and it is expected to see performance improvements when incorporating data with modality interference.

- Choice of the datasets: Why do the authors choose Mini-ImageNet and Caltech-101 as Image-heavy datasets and Open-BookQA and MMLU as text-heavy datasets? Would different choices of these datasets affect the performance of fine-tuned models on VQA tasks?

- Concern on the generalizability of the proposed method: While it is expected to see performance improvement on image-heavy and text-heavy datasets when they are incorporated into the fine-tuning dataset, it is unclear how this would mitigate modality interference on unseen image-heavy and text-heavy datasets.

- The proposed adversarial perturbation to the latent embeddings is not effective as compared to the perturbation in the input space as shown in Table 2. Is this component essential to mitigating modality interference?

**Questions:**

- How to choose image-heavy and text-heavy datasets? Would different choices of these datasets affect the performance of fine-tuned models on VQA tasks?
- Can the proposed method geeralize to unseen image-heavy and text-heavy datasets?
- Lines 264-265: How to choose $N_{img}$, $N_{text}$, and $N_{vqa}$ in practice to ensure effective modality interference mitigation?  How do different choices of these values affect the performance?

---

> ### Author Response · Authors · 2025-11-24
> **Rebuttal-part 1**
>
> We sincerely thank the reviewer for the constructive feedback.
>
> ## Technical Contributions
> We respectfully disagree that our method is merely "adding perturbed data." While heuristic augmentation injects noise, the critical difference lies in **Generalization vs. Memorization**, proven by convergent evidence from both our original OOD evaluations(table4,11) and new experiments on unseen datasets. As shown in results, Heuristic augmentation $D^{AUG}$ does not generalize: it memorizes specific perturbation patterns and fails on unseen datasets(Food101, ArcChallenge), often performing no better—or even worse—than the vanilla model. In contrast, our adversarial-invariance framework consistently improves robustness across both existing and entirely new domains, showing clear gains on real-world OCR/UI noise and strong transfer to unseen datasets. This demonstrates that our method learns true modality-level invariances rather than pattern-level augmentation effects.
>
> Technical novelty lies in: (1)**Structural Novelty**: we introduce a modality-specific causal mask to prevent perturbations from corrupting the task-relevant modality. Without this mask, standard global adversarial training ("No Mask") acts as naïve augmentation that degrades VQA reasoning (dropping from 63.8% to 61.5%). In contrast, our masked approach preserves semantic signals, serving as a "clean anchor" that improves accuracy to 66.8%; (2)**Algorithmic Novelty**: Unlike standard sign-based PGD, we employ raw-gradient updates tailored for embedding spaces to preserve semantic directions. This directed perturbation consistently outperforms Random Gaussian noise (e.g., 67.5% vs. 66.9% on LLaVA-1.5-13B VQA Overall), proving it captures the true interference mechanism rather than just adding data diversity; (3)**Systemic Novelty**: Unlike prior[1] works that apply uniform consistency in single-task settings, our framework introduces a task-aware, sample-specific consistency mechanism tailored for multi-task training. Since each batch dynamically mixes diverse tasks (image-heavy, text-heavy, VQA) requiring distinct perturbation strategies (Eq. 2), our method accurately tracks every origin sample and its variable number of perturbed variants to compute fine-grained consistency loss over answer sequences. This enables precise alignment under complex, heterogeneous task distributions with negligible additional computational cost.
>
> [1] Consistency Regularization for Cross-Lingual Fine-Tuning
>
> ## Choice of datasets
> Our selection of single-modality-heavy datasets is based on a simple principle: these datasets provide clean unimodal supervision, where one modality fully determines the label, allowing us to rigorously isolate and measure modality interference without the confounding factors present in naturally entangled VQA data.
>
> We agree that using different datasets would introduce different domain priors, and thus may shift the absolute VQA performance slightly, but the effectiveness of our mitigation mechanism remains invariant. However, the primary goal of this paper is **not to find the optimal combination of datasets** to maximize VQA scores, but to propose **a generalizable training framework** to mitigate modality interference and improve multimodal reasoning. We further emphasize that our training framework is **ataset-agnostic**. The method is built on a general causal principle—decoupling task-irrelevant modalities from the model’s reasoning process. Regardless of the dataset choices, the framework always constructs perturbed samples and enforces robustness through Consistency Regularization and Adversarial Perturbation, and experiments across all model families and scales show their effectiveness on top of the model fine-tuned with only image/text-heavy data.

---

> ### Author Response · Authors · 2025-11-24
> **Rebuttal-part 2**
>
> ## Generalize to unseen datasets
> |Model|Food101-Origin|Food101-MisleadText|Food101-OCRText|Arc-RandomPixels|Arc-RealImage|Arc-Screenshots|
> |-|-|-|-|-|-|-|
> |Qwen7b-Vanilla|97.1|96.8|93.3|96.7|85.6|79.6|
> |Qwen7b-FFT with $D^{AUG}$|97.1|96.6|94.1|85|82.9|80.2|
> |Qwen7b-Ours|97.5|97.6|97.4|97.4|84.1|82.6|
> |LLaVA7b-Vanilla|90.2|31|83.6|53.4|51.6|51.8|
> |LLaVA7b-FFT with $D^{AUG}$|91.3|40.2|83.5|55.7|52|53.8|
> |LLaVA7b-Ours|92.1|69|89.5|62.8|62|62.6|
>
> To rigorously assess generalizability, we extended our evaluation to Food-101 and ARC-Challenge, two datasets completely unseen during training and use the trained models for evaluation (training set consists only Caltech-101, Mini-ImageNet, OpenBookQA, MMLU and VQA datasets). Food-101[2] serves as an unseen image-heavy classification task. Under misleading text(seen perturb) and OCR-style(unseen perturb) noise, Vanilla models exhibited severe degradation, whereas our method preserved accuracy near clean-baseline levels. Similarly, on the ARC-Challenge[3] (text-heavy reasoning), our method consistently outperformed baselines under visual perturbations (Real Images, UI Screenshots). Meanwhile, heuristic data augmentation alone fails against unseen OOD noise and may even degrade performance.
>
> The results provide a definitive rebuttal to the claim that our method is just data augmentation. On Food-101 (OCR Noise), heuristic augmentation ($D^{AUG}$) completely fails to improve LLaVA-7B (83.5%) compared to Vanilla (83.6%). This proves that $D^{AUG}$ merely memorizes training patterns but fails to generalize to unseen noise types. In contrast, our method achieves 89.5% (+6.0%), and recovers performance on misleading text from 31.0% to 69.0%.
>
> ## Effect of adversarial training
> The results that input-space (heuristic) and latent (adversarial) perturbations show similar performance on seen tasks and perturbations(Table 2) are **expected**. While input-space perturbations(heuristic perturabtions) can help on some datasets, they only cover discrete, manually designed interference cases. Models can easily overfit to these heuristic patterns without learning a truly robust alignment mechanism. In contrast, our adversarial perturbation operates in continuous embedding space, searching for worst-case directions that input editing cannot expose. We do observe consistent performance gains in both VQA accuracy (table2, 7) and modality-interference mitigation (table 6) when incorporating adversarial training. However, the critical role of adversarial training is revealed in **Out-Of-Distribution scenarios**: as shown in Table 4, 11, and the results on new datasets, adversarial perturbations yields huge gains in OOD generalization.
>
> ## Batch constitution
> In practice, we adopt a 2:2:4 sampling ratio in a batch of 8 for image-heavy, text-heavy, and VQA data (Appendix D), which we empirically found to be the most stable configuration. Changing the ratio leads to predictable trade-offs. Increasing $N_{img}$/$N_{text}$ too much causes the model to forget cross-modal alignment, hurting VQA performance; increasing $N_{vqa}$ too much weakens the interference-mitigation signal, causing the model to behave similarly to the vanilla baseline. Imbalanced unimodal sampling (e.g., high $N_{img}$ but low $N_{text}$) yields asymmetric robustness—resistant to visual interference but still vulnerable to textual interference.
>
>
> [2] Food-101 – Mining Discriminative Componentswith Random Forests
>
> [3] Think you have Solved Question Answering? Try ARC, the AI2 Reasoning Challenge

---

> ### Author Response · Authors · 2025-11-26
>
> Dear Reviewer GRpQ,
>
> Thank you again for your detailed and thoughtful review. We’ve provided our rebuttal and would really appreciate any further feedback you may have. Looking forward to your thoughts and discussion!
>
> Sincerely,
>
> The Authors

---

### Official Review · Reviewer_WaXS · 2025-11-01

**Soundness:** 2
**Presentation:** 3
**Contribution:** 2
**Rating:** 4
**Confidence:** 3

**Summary:**

The paper first defines the Cross-Modality Competency Problem where existing Multimodal Large Language Models (MLLMs) are susceptible to misleading inputs especially in  modality-specific tasks, such as image classification or pure text question answering, where models are expected to rely solely on one modality. The paper, first benchmarks this across a range of models using a perturbation-based causal diagnostic setup. Perturbations include - for image-heavy tasks, unrelated scientific facts and misleading descriptions and - for text-heavy tasks are including a real, irrelevant image or a full black/white canvas image. Next, to improve upon this shortcoming, a novel framework to finetune MLLMs is presented which includes adversarial losses and a consistency regularizer strategy at the input and representation level.  Experiments on multiple datasets and models demonstrate significant improvements in robustness and cross-modality competency, indicating the method’s effectiveness in boosting unimodal reasoning ability while enhancing performance on multimodal tasks.

**Strengths:**

1. The definition of Modality Interference is well defined and the findings that model performance goes down due to sub-optimal integration information across modalities is interesting.
2. The motivation behind the proposed losses are well defined.
3. The paper is well written and easy to follow.

**Weaknesses:**

1. The proposed losses are not very effective : As shown in the ablations in Table 2, FFT with VQA/AUG performs better than proposed losses. Examples being : LLaVA-1.5-13B - FFT with $D^{AUG}$ - on ScienceQA-IMG | Qwen2.5-VL-3B - FFT with $D^{VQA}$ - on  MM-Bench-EN.
2. Consistency of results : In Table 1, the drop in performance of models on Caltech 101 is quite high, for example LLaVA-1.5-7B, goes from (97.0 --> 57.4), but in Table 4 : the drop is much less on OCR images  (97.0 --> 92.8) ; this raises questions around. i) if modality interference is really a concern and/or ii) if the generalization results in Table 4 are correct.
3. Provided results are on 3B/7B/13B models; results on newer family of thinking models would help solidify the claims made about failure modes in the paper.
4. To truly evaluate the effectiveness of the proposed losses, evaluation against other adversarial attacks should also be presented.

**Questions:**

1. It seems from Table 1 that drop in performance on models is much larger on vision-heavy tasks (such as Mini-ImageNet) is much larger than language-heavy tasks (such as OpenBookQA) - why might this be the case?
2. What is the reasoning behind choosing these perturbations :
i) unrelated scientific facts or misleading descriptions - for image-heavy tasks.
ii)  semantically meaningful real images/ full black canvas/ full white canvas - for language-heavy tasks.
3. Does the current evaluation of Modality Interference require Causal Modeling?
4. An ablation with/without the modality-specific binary mask for the adversarial loss will be interesting to see.

---

> ### Author Response · Authors · 2025-11-24
> **Rebuttal-Part 1**
>
> We sincerely thank the reviewer for the constructive feedback.
>
> ## Not Effective Solutions
>
> We thank the reviewer for their careful observation. While we acknowledge that FFT with $D^{AUG}$ or $D^{VQA}$ may occasionally outperform the full method on individual datasets, we respectfully argue that the proposed losses are very effective. The primary goal of our method is to achieve robust multimodal reasoning across both modality-heavy and multimodal tasks,  therefore our **primary evaluation metric*** is the **overall VQA performance** across datasets (Table 2,7) and the **degree of modality-interference reduction**(Table 3, 6, 11). Across all model families and all model scales, both consistency regularization and adversarail training consistently reduce modality interference, while the full method achieves the best overall VQA performance.
> While baselines like $D^{AUG}$ may occasionally outperform on individual heterogeneous datasets due to **task-specific overfitting**, our task-agnostic framework consistently achieves superior aggregate robustness and overall VQA performance across diverse architectures and settings.
>
> ## Consistency of Results
>
> We thank the reviewer for raising this point. The two drops correspond to **different perturbation categories** in our framework, as the large drop on Caltech-101 is caused by misleading descriptions (Table 5), where we intentionally add plausible but incorrect text (e.g., describing a “dog” for a cat image). This perturbation is designed to introduce semantic conflict into the irrelevant modality and therefore serves as a strong stress test; a large drop is expected.
> In contrast, the OCR snippets used in Table 4 are real-world OCR text (not images), which are unrelated-text that has no conflict with the image content. As shown in Table 5, unrelated facts only reduce Caltech-101 accuracy to $95.9\%$, closely matching the OCR results (97.0 → 92.8). This confirms that OCR noise belongs to the “irrelevant but non-conflicting” category and therefore produces only mild degradation.
>
> ## Evaluation on Thinking Models
>
> We agree that “thinking models” may have stronger potential to solve this problem by reasoning. To partially study this, we have already evaluated chain-of-thought prompting (Table 1). While CoT prompting reliably improves reasoning accuracy on clean inputs, it does not reduce modality interference, and the failure modes persist. We further conducted an evaluation using R1-Onevision-7B[1], a MLLM with enhanced reasoning capabilities. The results are:
> |Model|Caltech-101(Origin)|Caltech-101(MisleadText)|OpenBookQA(RandomPixels)|OpenBookQA(RealImage)|
> |-|-|-|-|-|
> |R1-OneVision-7b|96.5|84.7|81.2|77.6|
> |Qwen2.5-VL-7b|99.1|97.2|85.9|77.5|
>
> We investigated the generated reasoning traces: In some cases the model shows better awareness of irrelevant modalities, but in many cases the model still **treats unrelated modality as meaningful evidence and produces long hallucinated interpretations** that directly influence the final answer. This confirms that our methods remain necessary for interference mitigation.
>
> ## Evaluation on more Adversarial Attacks
>
> We additionally evaluate two representative adversarial attacks—FGSM[2] on text-heavy tasks (OpenBookQA) and GCG[3] on image-heavy tasks (Caltech-101). In FGSM, we adopt a perturbation bound of ε = 0.1. For GCG, we set top-K = 10 and use a 128-token adversarial suffix, following standard settings.
> |Model|Caltech-101(Origin)|Caltech-101(GCG K10L128)|OpenBookQA(RandomPixels)|OpenBookQA(FGSM ε 0.1)|
> |-|-|-|-|-|
> |LLaVA-1.5-7b|96.5|84.7|81.2|77.6|
> |Ours|99.1|97.2|85.9|77.5|
>
> Although adversarial robustness is not the goal of our work, these results show that our perturbation–consistency framework also strengthens resistance to standard adversarial attacks.
>
> ## Drop Difference in Image-heavy and Text-heavy tasks
>
> The difference in drop magnitude comes from the fact that the strongest perturbations in the two settings are **not symmetric**(details in table 5).
> For image-heavy tasks, the strongest perturbation-misleading description injects semantically conflicting text, creating a direct cross-modal contradiction. Such conflict is intentionally strong and therefore causes a large accuracy drop.
> For text-heavy tasks, the strongest perturbation is real images, which does not semantically contradict the question or answer. These perturbations function as non-conflicting noise.
> Therefore, comparing absolute drop sizes across these qualitatively different perturbations is not meaningful; the perturbations **differ in causal strength** rather than reflecting inconsistency in our results.
>
> [1] R1-Onevision: Advancing Generalized Multimodal Reasoning through Cross-Modal Formalization
>
> [2] Explaining and Harnessing Adversarial Examples
>
> [3] Universal and Transferable Adversarial Attacks on Aligned Language Models

---

> ### Author Response · Authors · 2025-11-24
> **Rebuttal-part 2**
>
> ## Reasons behind perturbations
> Our perturbations are systematically chosen to reflect a spectrum of **causal strengths** when intervening on the irrelevant modality:
>
> - Image-Heavy Tasks: *Unrelated facts* serve as neutral distractions, while *misleading descriptions* introduce strong semantic conflict (actively contradicting the visual content).
>
> - Text-Heavy Tasks: *Black/white canvases* serve as neutral visual noise, whereas *real images* provide semantically meaningful but irrelevant interference, testing the model's ability to suppress rich visual features that are contextually invalid.
>
> These interventions are intentionally simple yet diagnostic. Crucially, while our evaluation employs these specific heuristics, our mitigation method is **perturbation-agnostic**. Adversarial embedding perturbations operate in continuous space and generalize strongly to unseen real-world corruptions (Table 4, 11), indicating that our framework learns a general mechanism for cross-modality competency rather than overfitting to specific perturbation patterns.
>
> ## Causal Modeling
> Causal modeling is needed because Modality Interference is fundamentally a causal phenomenon rather than a generic robustness issue.
>
> **(1)Formalizing why the problem is causal**: Rather than viewing interference as generic robustness degradation, causal framing provides a clear definition: interference occurs when changing the irrelevant modality alters the model’s prediction. This prevents conflating our phenomenon with ordinary distribution shift or noise sensitivity.
>
> **(2)Quantifying the Total Causal Effect**: The prediction change rate $\delta_{cp}$ is a direct empirical estimation of the Total Causal Effect (TCE) of the irrelevant modality. It quantifies how much the prediction shifts when only that modality is intervened on, giving a clean measure of interference strength without confounding factors.
>
> **(3)Guiding the Mitigation approach**:  As noted in the paper, this causal evaluation serves as the necessary diagnostic entry point. A large TCE reveals an active spurious pathway, which motivates our mitigation apporach: we move from observing the total effect (TCE) to control the Direct Causal Effect (DCE) of latent representations ($Z_I, Z_T$) via adversarial training and consistency regularization.
>
> ## Ablation without Modality Mask
> We thank the reviewer for proposing this insightful ablation. We implemented the "No Mask" variant, where adversarial perturbations are applied to all modalities across all datasets (including VQA).
>
> |Model|Caltech-101(origin)|Caltech-101(MisleadText)|Caltech-101(OCRText)|OpenbookQA(RandomPixel)|OpenbookQA(RealImage)|OpenbookQA(Screenshots)|VQAOverall|
> |-|-|-|-|-|-|-|-|
> |llava7b-vanilla|97|57.4|92.8|62.4|56.4|50.2|63.8|
> |llava7b-Ours(No Mask)|98.1|96.3|93.5|79.5|78.1|62.7|61.5|
> |llava7b-Ours|99.3|98.9|99|81.8|81|81.8|66.8|
>
> As shown in table, (1)no mask training still partially mitigates modality interference—because the combination of consistency regularization and simple heuristic perturbations already teaches the model useful invariances, and adversarial training itself improves generic robustness(e.g. Screenshots 50.2 → 62.7). (2)The mask is crucial for **OOD generalization**: No mask leads to collapses on unseen perturbations (e.g., Screenshots: 81.8 → 62.7). (3) The mask prevents **semantic destruction** during VQA training: when all the token embeddings are perturbed, the model cannot preserve multimodal semantics, leading to a notable drop in VQA accuracy (66.8 vs. No Mask 61.5).

---

> ### Author Response · Authors · 2025-11-26
>
> Dear Reviewer WaXS,
>
> Thank you again for your detailed and thoughtful review. We’ve provided our rebuttal and would really appreciate any further feedback you may have. Looking forward to your thoughts and discussion!
>
> Sincerely,
>
> The Authors

---

> > ### Comment · Reviewer_WaXS · 2025-11-27
> >
> > Thanks for your rebuttal. I have 2 main follow-up questions :
> >
> > 1. On Effectivness of Losses -- I am not sure if the authors are fully clear on the motivation of the work. Figure 1, which I believe is the motivating figure, shows results on Image-Heavy and Text-Heavy tasks but in their response mention that their primary evaluation metric is the overall VQA performance. This seems to be a contradiction. In any case, if we only look at the VQA performance in Table 2, the gains are actually quite marginal.
> >
> > 2. Is there a scientific intuition as to why these perturbations are chosen? Specifically, if instead of using unrelated 'scientific' facts, we use unrelated facts from Domain A, B ... etc. -- will these findings still hold? Also, the authors use OCR, UI data as distractors, is there a specific reason why? Are there some domains where the images dont cause as much a performance drop?

---

> > > ### Author Response · Authors · 2025-11-28
> > >
> > > Thank you very much for these helpful follow-up questions. We appreciate the opportunity to clarify both points, and we will incorporate the requested explanations into the revised manuscript.
> > >
> > > ## On Effectivness of Losses
> > > We understand the concern regarding the two evaluation modes, and we apologize if the current presentation made them appear contradictory. We want to clarify that there is no contradiction; our evaluation reflects the dual nature of Cross-Modality Competency-the model's inability to **fairly evaluate all modalities**. Therefore, a complete solution must demonstrate two capabilities(**we have clearly stated in both the general response and the initial rebuttal**): (1)Robustness (The Symptom): The ability to ignore irrelevant modalities when they act as noise. This is evaluated via Interference Reduction (Table 1, 3, 6, 11). (2)Reasoning (The Root Cause): The ability to utilize relevant modalities correctly when required. This is evaluated via Overall VQA (Table 1, 2, 7).
> > >
> > > Regarding Table 2, we do acknowledge that the gains may appear modest when compared against heuristic augmentations($D^{AUG}$). However, against the vinilla model and the VQA fine-tuning baselines ($D^{VQA}$), the improvements are statistically meaningful. Importantly, in adversarial learning literature, improving robustness typically incurs an "alignment tax"—a drop in clean accuracy [4]. The fact that our method **achieves massive robustness gains** while **simultaneously improving clean VQA performance** is a non-trivial achievement. We will make this context explicit to avoid misinterpretation.
> > >
> > > [4] Robustness May Be at Odds with Accuracy, ICLR 2019
> > >
> > > ## Scientific Intuition and Perturbation Choices
> > >
> > > We appreciate your question about the intuition behind choosing these perturbations. Our primary objective is to **prove the existence of modality interference** and **provide a robust mitigation strategy**, rather than exhaustively cataloging every possible noise type. We agree that clarification is needed and will revise the text to make this motivation clearer.
> > >
> > > We chose these specific perturbations as canonical proxies for two fundamental causal categories:
> > > + Semantic Conflict (e.g., Misleading Descriptions): Testing resistance to contradiction.
> > > + Semantic Irrelevance (e.g., Unrelated Facts): Testing resistance to distraction. The findings and our mitigation method will still hold for domain-specific facts. As long as the text is semantically unrelated (breaking the causal link), the specific domain does not alter the mechanism of interference. Meanwhile, our proposed method is perturbation-agnostic as stated in the general response.
> > >
> > > We selected OCR snippets and UI screenshots specifically to bridge the gap between Diagnostic Control and Real-World Utility for two key reasons: (1)**Realistic Usage Scenarios**: This choice reflects how humans actually interact with MLLMs "in the wild." In real-world deployment, users frequently upload images containing incidental noise—such as scanned documents with dense text (OCR) or mobile screenshots with complex UI elements—while asking unrelated questions. A robust model must be able to ignore these naturally occurring visual/textual distractors, not just synthetic noise. (2)**Unseen OOD Validation**: As these noise types were never seen during training, they serve as a rigorous Out-Of-Distribution (OOD) test. Success on these perturbations (Table 4/11) confirms that our method generalizes to novel, real-world noise distributions, rather than merely overfitting to the specific training templates.
> > >
> > > Finally, we appreciate your question about domain-specific effects. While we did not conduct a fine-grained sweep across specific domains, we effectively investigated this by comparing inputs with different levels of semantic density. Our results (Table 8) reveal that Semantic Density drives the drop magnitude. High-density inputs (Real Images) trigger strong interference, whereas low-density inputs (Black Canvas/Noise) cause minimal drops. This implies that domains lacking salient semantic features trigger less interference. We will highlight this more clearly and discuss domain sweeps as an interesting direction for future work.

---

### Official Review · Reviewer_MwUa · 2025-11-01

**Soundness:** 3
**Presentation:** 3
**Contribution:** 4
**Rating:** 6
**Confidence:** 3

**Summary:**

The paper investigates modality interference in MLLMs, particularly in tasks like Visual Question Answering. It defines modality interference as the negative impact of irrelevant modalities on unimodal tasks and quantifies this issue through perturbation-based causal diagnostics.

To mitigate this interference, the authors introduce a new fine-tuning framework that incorporates data augmentation and consistency regularization strategies to improve model stability across different inputs. Experimental results demonstrate significant enhancements in robustness and overall performance.

**Strengths:**

Innovative Concept: The paper introduces the notion of the Cross-Modality Competency Problem, providing a fresh perspective on modality interference in multimodal large language models. This innovative approach contributes new insights to the field.

Systematic Analysis: By designing a perturbation-based causal diagnostic experiment, the authors quantify the impact of modality interference, providing empirical evidence that enhances the scientific rigor and validity of the research.

Effective Solution: The proposed fine-tuning framework combines data augmentation with consistency regularization strategies, offering a practical solution to mitigate modality interference. This approach has been validated through significant improvements in robustness and performance across multiple benchmark datasets.

**Weaknesses:**

Potential Overfitting Risks: The use of perturbation-based data augmentation may introduce noise into the training process. While it aims to enhance robustness, there is a risk that the model might overfit to these perturbed examples, resulting in poorer generalization on clean, real-world data.


Lack of Comparative Baselines: The paper does not provide a comprehensive comparison against a wider variety of existing methods or models that address modality interference. Without robust baseline comparisons, it is difficult to ascertain the relative effectiveness of the proposed framework.


Limited Experimental Diversity: The experiments primarily focus on a small set of benchmark datasets, which may not capture the full range of conditions under which modality interference could occur. This limited range could restrict the generalizability of the findings to other real-world scenarios.

**Questions:**

How does the performance of the model vary with different configurations of the augmentations or regularization strategies?

---

> ### Author Response · Authors · 2025-11-24
>
> We sincerely thank the reviewer for the constructive feedback.
>
> ## Potential Overfitting Risks
> We would like to clarify that our perturbation-based training does not encourage the model to memorize noisy inputs, but instead explicitly aims to make the model ignore irrelevant-modal variations.
> First, Table 4 provides direct empirical evidence against overfitting. When evaluated on real-world perturbations—such as noisy OCR text for image-heavy tasks and unrelated screenshots for text-heavy tasks—models trained with our method consistently outperform the perturbation-data trained model setting (FFT with $D^{AUG}$ )while preserving performance on clean datasets. Additional experiments on new datasets (food101, arc challenge) further enhance this claim. This demonstrates that the model does not overfit to synthetic perturbations; instead, it generalizes better to naturally occurring cross-modal noise, precisely the failure mode our method aims to mitigate.
> Second, our training objective discourages fitting any perturbation-specific patterns. The consistency regularization and adversarial perturbation act as invariance-inducing mechanisms: they force the model to produce stable predictions under interventions on the irrelevant modality. This is aligned with adversarial debiasing [1] and invariant learning [2], which are widely shown to reduce overfitting rather than cause it.
>
> ## Lack of Comparative Baselines
>
> We utilized a representative selection of baselines because Modality Interference is a newly formalized problem with limited direct predecessors. To ensure fairness, we selected baselines that represent three strongest and most relevant paradigms: (1)Architectural approaches: I-MoF (NeurIPS 2024) improves cross-modal fusion. (2)Data-centric tuning: VLMClassifier (CVPR 2024) restores unimodal grounding. (3)Inference prompting: CoT tests whether reasoning alone can mitigate interference.
>
> These baselines cover the major existing directions for addressing cross-modal degradation, and our method consistently outperforms them across all settings.
>
> ## Limited Experimental Diversity
>
> We emphasize that our initial dataset selection (canonical image/text-heavy tasks) was deliberate to strictly isolate unimodal signals for causal diagnosis, rather than for exhaustive benchmarking. To address the concern regarding experimental diversity and generalizability, we have extended our evaluation to (1)Real-World OOD Robustness:  As shown in Table 4 and 11, our method generalizes to real world noise, such as noisy OCR snippets and unrelated UI screenshots, improving robustness while preserving clean accuracy. (2)unseen datasets(Food 101 for image-heavy, ARC-Challenge for text-heavy with results in general response): Our methods also generalize well on new datasets.
>
> Both Modality interference and our mitigation solution are **architecture- and dataset-agnostic**. Across multiple MLLM families, multiple model sizes, and multiple task regimes (image-heavy, text-heavy, VQA) under both ID and OOD evaluation, we consistently observe the same failure mode and the same improvements from our mitigation method. This cross-architecture consistency demonstrates that modality interference is not tied to any specific dataset, but reflects a general property of multimodal LLM inference.
>
>
> ## Sensitivity Analysis
>
> We have performed sensitivity analysis on adversarial training iterations. As shown in Appendix Fig. 5, varying the number of iterations (1–5 steps) produces consistent interference-mitigation improvements across both 7B and 13B model sizes. This indicates that the effectiveness of our method does not depend on a specific perturbation-strength setting.
> We also note that small fluctuations across iteration counts are fully expected. Prior work [1, 3] on adversarially-perturbed representations reports similar non-monotonic patterns when varying iterations, indicating the intrinsic sensitivity of adversarial optimization rather than instability of our method. Furthermore, as discussed in Appendix E, more iterations increase computational and wall-clock cost, and adopting two iterations provides the best balance of mitigation effectiveness, stability, and efficiency. This motivates our default configuration.
> We additionally sweep the KL weight on llava-1.5-7b. As shown in table, The method is insensitive to the specific magnitude, maintaining robust performance across a wide range ($0.001 \le \lambda \le 0.05$).
>
> |Lambda|VQA-Overall|
> |---|---|
> |0|65.55|
> |0.001|66.2|
> |0.005|66.48|
> |0.01|66.81|
> |0.05|66.63|
>
> [1] Towards Deep Learning Models Resistant to Adversarial Attacks.
>
> [2] Virtual Adversarial Training: A Regularization Method for Supervised and Semi-Supervised Learning
>
> [3] PTP: Boosting Stability and Performance of Prompt Tuning with Perturbation-Based Regularizer

---

> ### Author Response · Authors · 2025-11-26
>
> Dear Reviewer MuUa,
>
> Thank you again for your detailed and thoughtful review. We’ve provided our rebuttal and would really appreciate any further feedback you may have. Looking forward to your thoughts and discussion!
>
> Sincerely,
>
> The Authors

---

### Author Response · Authors · 2025-11-24

**General Response to All Reviewers**

We sincerely thank all reviewers for their constructive feedback and insightful suggestions. Before addressing specific questions in individual responses, we concisely summarize our contributions and clarify key aspects regarding our research goals, the critical role of adversarial training, and the generalizability of our framework.

### 1. Summary of Contributions & Novelty
We are the first to define, analyze, and address **Modality Interference** as a concrete manifestation of the broader **Cross-Modality Competency Problem**, moving from ad-hoc fixes to a principled, generalizable solution.

* **Problem:** Current MLLMs fail to fairly evaluate information across modalities. In unimodal tasks, spurious cues from irrelevant modalities (e.g., misleading text in image tasks) significantly degrade performance (Figure 1).
* **Challenge:** Existing methods target narrow symptoms (e.g., catastrophic forgetting) or trade off unimodal accuracy for multimodal performance. There is a lack of end-to-end approaches for both **diagnosis** and **mitigation**.
* **Insight:** We identify modality interference as a measurable causal phenomenon. Our Causal Graph formalization allows us to quantify interference strength via the Total Causal Effect.
* **Solution:** A unified framework grounded in causal analysis.
    *  *Diagnosis:* Causal interventions quantify modality interference and reveal how irrelevant signals disrupt reasoning..
    *  *Mitigation:* Fine-tuning with heuristic & adversarial perturbation-based augmentation and consistency regularization constrains the model to follow task-relevant causal paths, improving robustness. This constrains the model to sever spurious shortcuts and follow task-relevant causal paths.
* **Results:** Across three architectures and multiple scales, our method consistently achieves **Pareto-optimality**: boosting unimodal robustness while enhancing general multimodal reasoning.

### 2. Clarification on Research Goals: Robustness AND Reasoning
We emphasize that our objective is **twofold**: to mitigate modality interference *and* to elevate overall multimodal reasoning capabilities.

Our results demonstrate that these goals are not mutually exclusive. By suppressing "easy" non-causal shortcuts, our method compels the model to learn robust, semantic cross-modal alignment. This directly translates to improved performance on complex VQA benchmarks, proving that restoring cross-modality competency enhances the model's fundamental reasoning ability.

### 3. The Critical Role of Adversarial Training & New OOD Experiments
A key concern raised was the necessity of Adversarial Training (ADV) versus simple data augmentation. We clarify that ADV is **imperative for Out-Of-Distribution (OOD) Generalization**.

To validate this, we extended our evaluation to **completely unseen datasets** (Food-101 and ARC-Challenge, new added experiments) and **real-world noise** (OCR snippets, UI Screenshots, listed in table 4 and 11) that were never seen during training.

* **Heuristic Augmentation Fails:** Simple augmentation ($D^{AUG}$) acts as memorization. It fails to generalize to these unseen distributions.
* **Adversarial Training Generalizes:** Our ADV-based framework yields massive gains on these unseen tasks.

This confirms that our method is **perturbation-agnostic** and **dataset-agnostic**. The adversarial component searches for continuous "worst-case" directions in the embedding space in training time, enabling the model to learn true modality invariance and fairly to evaluate all modalities for cross modal competency that transfers to novel scenarios.

We will incorporate all new experiments and discussions into the revised manuscript. We thank the reviewers again for helping us strengthen the comprehensiveness and theoretical depth of this work.

---

### Meta-Review · Area_Chair_MkmC · 2026-01-03

**Summary:**

The submission initially received mixed reviews (4466). The main concerns can be summarized into the following points:
1. The theoretical depth and technical contributions are limited. It largely integrates known ideas from causal inference and adversarial robustness. More insights are expected.
2. More comprehensive evaluations are needed. For example, 1) a comprehensive comparison against existing methods; 2) more benchmark datasets are expected. 3) generalization ability of the proposed method; 4) efficiency trade-off needs to be discussed.
3. The proposed method is not very effective, or the improvements are marginal.
4. More explanations and motivations about some methodology designs are necessary, such as the reason behind the choice of different perturbations, the reason behind the choice of the datasets.

In the rebuttal, the authors have provided extra experimental results to address the second concern. But the first and third concerns are not well-addressed. Thus, I recommend Reject.

**Reviewer Concerns:**

The authors have addressed the concerns about more comprehensive evaluations, but the presetnation of its novelty/contributions/insights and marginal gains are not well addressed.

**Reviewer Scores:**

I think all reviewers will not change their ratings after the rebuttal.

---

### Decision · Program_Chairs · 2026-01-26

Reject